# Two complementary genes in a presence-absence variation contribute to *indica-japonica* reproductive isolation in rice

Daiqi Wang[1,2,3,4], Hongru Wang[5], Xiaomei Xu[1], Man Wang[1,2], Yahuan Wang[1], Hong Chen[1], Fei Ping[1,2], Huanhuan Zhong[1], Zhengkun Mu[1,2], Wantong Xie[1,2], Xiangyu Li[1,2], Jingbin Feng[1,2], Milan Zhang[1,2], Zhilan Fan[6], Tifeng Yang[7], Junliang Zhao [1,7], Bin Liu[7], Ying Ruan[3,8], Guiquan Zhang[1,2], Chunlin Liu[3,4] & Ziqiang Liu [1,2] ✉

Understanding the evolutionary forces in speciation is a central goal in evolutionary biology. Asian cultivated rice has two subspecies, *indica* and *japonica*, but the underlying mechanism of the partial reproductive isolation between them remains obscure. Here we show a presence-absence variation (PAV) at the *Se* locus functions as an *indica-japonica* reproductive barrier by causing hybrid sterility (HS) in *indica-japonica* crosses. The locus comprises two adjacent genes: *ORF3* encodes a sporophytic pollen killer, whereas *ORF4* protects pollen in a gametophytic manner. In $F_1$ of *indica-japonica* crosses, pollen with the *japonica* haplotype, which lacks the sequence containing the protective *ORF4*, is aborted due to the pollen-killing effect of *ORF3* from *indica*. Evolutionary analysis suggests *ORF3* is a gene associated with the Asian cultivated rice species complex, and the PAV has contributed to the reproductive isolation between the two subspecies of Asian cultivated rice. Our analyses provide perspectives on rice inter-subspecies post-zygotic isolation, and will promote efforts to overcome reproductive barriers in *indica-japonica* hybrid rice breeding.

Detection and characterization of structural variants (SVs) has revolutionized the understanding of the landscape of genomic and phenotypic variation in different species[1,2]. Presence-absence variation (PAV) is an extreme form of SV, in which a genomic segment containing one or more genes is present in some individuals but absent in others[3]. The unprecedented low cost of DNA sequencing and advances in high-quality genome assembly today make it possible to generate a wealth of data on PAVs in plant genomes[4–13]. Pangenomic and gene ontology analyses in crops consistently find that PAVs are enriched for abiotic stress and disease-response genes[4,5,9,10,13]. Increasing evidence has confirmed that PAV is one of the genetic factors underlying variation for agronomic traits such as submergence tolerance (conferred

[1]Guangdong Provincial Key Laboratory of Plant Molecular Breeding, College of Agriculture, South China Agricultural University, Guangzhou, Guangdong 510642, China. [2]State Key Laboratory for Conservation and Utilization of Subtropical Agro-Bioresources, South China Agricultural University, Guangzhou, Guangdong 510642, China. [3]Key Laboratory of Hunan Provincial on Crop Epigenetic Regulation and Development, Hunan Agricultural University, Changsha, Hunan 410128, China. [4]College of Agronomy, Hunan Agricultural University, Changsha, Hunan 410128, China. [5]Shenzhen Branch, Guangdong Laboratory of Lingnan Modern Agriculture, Genome Analysis Laboratory of the Ministry of Agriculture and Rural Affairs, Agricultural Genomic Insitute at Shenzhen, Chinese Academy of Agricultural Sciences, Shenzhen, Guangdong 518120, China. [6]National Field Genebank for Wild Rice (Guangzhou), Rice Research Institute, Guangdong Academy of Agricultural Sciences, Guangzhou, Guangdong 510640, China. [7]Guangdong Provincial Key Laboratory of New Technology in Rice Breeding, Rice Research Institute, Guangdong Academy of Agricultural Sciences, Guangzhou, Guangdong 510640, China. [8]College of Bioscience and Biotechnology, Hunan Agricultural University, Changsha, Hunan 410128, China. ✉e-mail: zqliu@scau.edu.cn

by *Sub1A*[14], *SNORKEL1* and *SNORKEL2*[15]), phosphous-deficiency tolerance (conferred by *Pstol*[16]), anaerobic germination tolerance (conferred by *OsTPP7*[17]), disease resistance (conferred by *LRK1*[18] and *OsWAK112d*[6]), and plant architecture (conferred by *SLB1* and *SLB2*[19]). In addition, fixation of complementary PAVs is believed to contribute to heterosis in hybrid breeding[20].

Asian cultivated rice (*O. sativa* L.), which comprises the staple food for half of the global population, was domesticated from common wild rice (*O. rufipogon* Griff.) in Asia, and has two subspecies, *O. sativa* L. ssp. *indica* and *O. sativa* L. ssp. *japonica* (hereafter referred as *indica* and *japonica*, respectively)[21]. As two genetically diverged subspecies, *indica* and *japonica* varieties display distinct morphological, physiological and biochemical traits, reflecting the adaptive evolution of the two subspecies in divergent natural and human-influenced environments.

Various types of post-reproductive barriers exist between *indica* and *japonica* rice accessions[22]. Hybrid sterility (HS) is the most commonly found reproductive barrier in *indica-japonica* crosses, and is often cited as the criterion to classify *indica-japonica* subspecies[23]. Molecular genetic studies over two decades have revealed three mechanisms of HS in *Oryza*, which have been designated as the killer-target system, the killer-protector system, and the duplicate gametic lethal system[24]. These involve 9 single-locus HS loci (*Sa*[25], *Sc*[26], *qHMS7*[27], *S5*[28], *S7*[29], *HSA1*[30], *ESA1*[31], *S1*[32,33], *pf12A*[34]) and 3 two-locus HS pairs (*DPL1/DPL2*[35], *S27/S28*[36], *DSG1/DSG2*[37]). Despite extensive studies of these loci, we still lack efficient methods to fully overcome HS in inter-specific and inter-subspecific hybrid rice, and our understanding of the role of HS in genome evolution remains limited.

The yield potential of *indica-japonica* hybrid rice is estimated to be 15–30% higher than that of the *indica* intraspecific hybrid rice widely cultivated today. The utilization of strong heterosis in *indica-japonica* hybrids is one of the most promising future directions in rice breeding[24,38–40], therefore, understanding *indica-japonica* reproductive isolation is important to unlock their breeding potential. *Se/pf12* is a major locus conferring hybrid male sterility in *indica-japonica* hybrids, which was identified by a large number of studies using different populations[34,41–47]. Artificial disruption of *pf12A* gene at *Se/pf12* locus successfully converts this hybrid sterility locus into a neutral allele[34]. However, the molecular mechanism of *pf12A* regulating rice HS remains elusive.

In this work, we show that a PAV at the *Se* locus functions as an *indica-japonica* reproductive barrier by causing HS in *indica-japonica* crosses. The complementary effect of two adjacent genes at *Se* locus within the PAV region induces hybrid male sterility (HMS), and thus reproductive isolation between *indica* and *japonica*. Evolutionary analyses of *Se* locus show that the PAV is an important contributor to the postzygotic isolation between the two subspecies, and it also supports the independent domestication of *indica* and *japonica* from different *O. rufipogon* populations. Based on these findings, effective approaches could be designed to break down reproductive barriers in *indica-japonica* hybrid rice breeding.

## Results

### Identification of *Se* controlling HMS
In order to identify genes responsible for *indica-japonica* HS, near isogenic lines (NILs) were developed using *japonica* variety T65 as the recurrent parent and *indica* variety GLA4 as the donor parent. In the BC$_5$F$_2$ generation, a NIL, E9, was identified that had four substituted segments in the genetic background of T65 (Fig. 1a and Supplementary Fig. 1). T65, T65/E9 F$_1$ hybrids and E9 exhibited similar plant morphology except for anther length (Fig. 1b and Supplementary Figs. 2 and 3a–c). Pollen staining using both I$_2$-KI and TTC showed that male fertility was normal for T65 and E9; however, the T65/E9 F$_1$ hybrid was semi-sterile, with a mix of fertile and defective pollen (Fig. 1c, d). Pollen grains of T65 and E9 were larger and spherical (Supplementary Fig. 3d,

f–h), whereas T65/E9 F$_1$ hybrids produced two kinds of pollen grains: normal, fertile pollen filled with abundant starch granules, and defective pollen filled with cytoplasm but without any starch granules (Supplementary Figs. 3e, i, j and 4a, b). Defective pollen showed thicker exines and intines than fertile pollen, suggesting that pollen wall formation was also affected in T65/E9 F$_1$ hybrids (Supplementary Fig. 4c–e). Semi-thin sections of developing anthers indicated that half of the microspores were aborted in T65/E9 F$_1$ hybrids starting from the bicellular pollen stage (St11) (Supplementary Fig. 5). At the tricellular pollen stage (St12), the aborted pollen resembled St10 vacuolated microspores, whereas normal mature pollen grains were densely stained and filled with starch granules (Supplementary Fig. 5).

Segregation analysis of 167 T65/GLA4 BC$_5$F$_2$ plants during E9 construction revealed severely distorted segregation of two SSR markers (RM19 and RM453) on chromosome 12, with significantly lower than the expected 25% homozygous T65 genotype, while SSR markers on chromosome 2, 3 and 7 showed normal 1:2:1 mendelian segregation (Supplementary Table 1). Test crosses with T65/E9 F$_1$ hybrids as the male parent and T65 or E9 as female parents indicated that the male gametes derived from T65 were selectively aborted and T65 alleles were not transmitted to the progeny (Supplementary Table 2). Conversely, test cross with T65/E9 F$_1$ hybrids as the female parent and E9 as the male parent presented 54:74 segregation ratio of *T/E* and *E/E* (Supplementary Table 2), implying that the female gametes carrying T65 allele is also affected, but at a relatively lower degree compared to male gametes, which is consistent with slightly lower spikelet fertility for T65/E9 F$_1$ hybrid (Fig. 1e).

For 279 plants of T65/E9 F$_3$ segregating population, spikelet fertility showed a normal distribution; however, a clear bimodal pattern with an apparent valley at 70–80% was observed for pollen fertility (Fig. 2a, b). Taking 70–80% as the cut-off region between semi-sterility and fertility, the population segregated for semi-sterility (<70%): fertility (>80%) at the ratio of 1:1 ($\chi^2 = 0.13 < \chi^2_{0.05,1} = 3.84$), demonstrating monogenic inheritance for the trait. Our results indicated the substituted segment on chromosome 12 contained a gene locus, designated *Se* as in previous studies[41,42], controlling HMS between T65 and E9, which was closely linked with RM19 and RM453 and appeared to be allelic to several previously identified HMS QTLs[34,43–47].

### Complex genomic structural variations at *Se* locus
Linkage analysis of pollen fertility and SSR markers using the above 279 T65/E9 F$_3$ plants pinpointed the *Se* locus to an interval between IND9 and PSM448 (Fig. 2c). Through analysis of 7,274 F$_4$ plants, *Se* was further restricted to the genomic region between markers PSM623 and IND48 (Fig. 2d and Supplementary Table 3). Sequence analysis revealed two genes (*ORF1*$^{T65}$ and *ORF2*$^{T65}$) in the 13.17-kb *Se* region in T65 (Fig. 2e). *ORF1*$^{T65}$ is predicted to encode a 250-aa protein without any conserved domains and *ORF2*$^{T65}$ is predicted to encode a 466-aa alpha/beta hydrolase family protein. Sequence comparison revealed nucleotide diversity for *ORF1* and *ORF2* between T65 and E9. Compared with *ORF1*$^{E9}$, a 378-bp deletion in the last exon and 11 single-nucleotide polymorphisms (SNPs) resulted in a 126-aa deletion and 5 aa substitutions, respectively, in ORF1$^{T65}$ (Supplementary Fig. 6). Although several indels and SNPs exist between *ORF2*$^{T65}$ and *ORF2*$^{E9}$, causing premature termination in the predicted ORF2$^{E9}$ (Supplementary Fig. 7), the major difference in *ORF2* is in the promoter region, which is completely different between the two alleles. *ORF2*$^{E9}$ expression could not be detected in anthers or in any other tested tissues (Supplementary Fig. 8). In E9, a 28.45-kb insertion was found between *ORF1*$^{E9}$ and the putative *ORF2*$^{E9}$, containing two additional genes, *ORF3* and *ORF4* (Fig. 2e). *ORF3* is predicted to encode a 596-aa superfamily II DNA and RNA helicase with a mitochondrial targeting signal (Supplementary Fig. 9), and *ORF4* is predicted to encode a 523-aa protein with ankyrin repeats (Supplementary Fig. 10).

### *ORF3* encodes a sporophytic pollen killer

In order to identify the genes responsible for HMS, we used the CRISPR/Cas9 genome editing system to knock out T65 candidate genes in T65 or E9 candidate genes in E9, respectively. Unique target sites were selected in the coding region of each gene to make sure loss-of-function mutations were obtained. Three *orf3* homozygous mutants (*orf3-1* to *orf3-3*) and two homozygous mutants each for *orf1*^T65, *orf2*^T65, *orf1*^E9 and *orf1*^T65*orf2*^T65 with different editing patterns, respectively, were selected and showed normal pollen viability, suggesting that *ORF1*, *ORF2* and *ORF3* are not essential for the development of male gametophytes (Fig. 3a, Supplementary Fig. 11a−d and Supplementary Tables 4, 5). The mutants were further crossed with T65 or E9 to produce mutant-type *Se* heterozygotes to test HMS. In contrast to the semi-sterile pollen exhibited by T65/E9 $F_1$ hybrids and mutant-type $F_1$ hybrids with *orf1*^T65, *orf2*^T65, *orf1*^E9 or *orf1*^T65*orf2*^T65 alleles (Supplementary Fig. 11e−h and Supplementary Table 4), mutant-type $F_1$ hybrids with the three *orf3* alleles exhibited normal pollen fertility (Fig. 3b). Moreover, the $F_2$ families showed the normal 1:2:1 segregation ratio, indicating that loss-of-function of *ORF3* restores male fertility in *Se* heterozygotes and eliminates the segregation distortion (Fig. 3c). These results are in consistent with normal pollen fertility and 1:2:1 segragation of two heterozygous *pf12A* (allelic to *ORF3*) mutants[34]. To further study *ORF3*'s function, a genomic fragment containing *ORF3* or *ORF3* overexpression constructs were introduced into T65. In total, 34 out of 41 (genomic *ORF3*), and 3 out of 28 (*ORF3* overexpression) transgenic lines contained detectable transgenes. Surprisingly, both

the transgene-positive lines and transgene-negative lines showed normal pollen fertility (Supplementary Table 6). Expression analysis revealed that *ORF3* expression in the transgene-positive lines was very low compared with that in T65/E9 $F_1$ hybrids (Supplementary Fig. 12), suggesting that *ORF3* is toxic to living cells without *ORF4* and its expression should be strictly regulated under T65 genetic background. Direct evidence of *ORF3*'s killing effect was obtained from the transgenic callus experiment in which the selection frequency of Hyg-resistant callus was extremely low for *ORF3* compared with the empty vector control (Supplementary Fig. 13a, b), and transgene-positive calli showed low expression of transgene (Supplementary Fig. 13c), indicating that the growth of T65 callus was retarded or inhibited when *ORF3* was introduced. Similarly, the induction of ORF3 in yeast significantly inhibited the growth of the transformants (Supplementary Fig. 14). Our results indicated that *ORF3* has a pollen-killing function and acts in a sporophytic manner.

### *ORF4* performs a pollen-protection function in a gametophytic manner

The selective abortion of T65-type pollen in T65/E9 $F_1$ hybrids but not in T65 suggests that a pollen protector is most likely encoded in E9 to eliminate the pollen-killing effect of *ORF3*. To test this notion, we transformed intact *ORF1*^E9, *ORF2*^E9 and *ORF4* genomic fragments from E9 into T65, respectively. Hemizygous transgenic $T_0$ plants were further crossed with E9 to produce $F_1$ hybrids carrying the pollen killer *ORF3* (Fig. 4a). For $F_1$ plants carrying the *ORF4* transgene (*T/E; ORF4/-*),

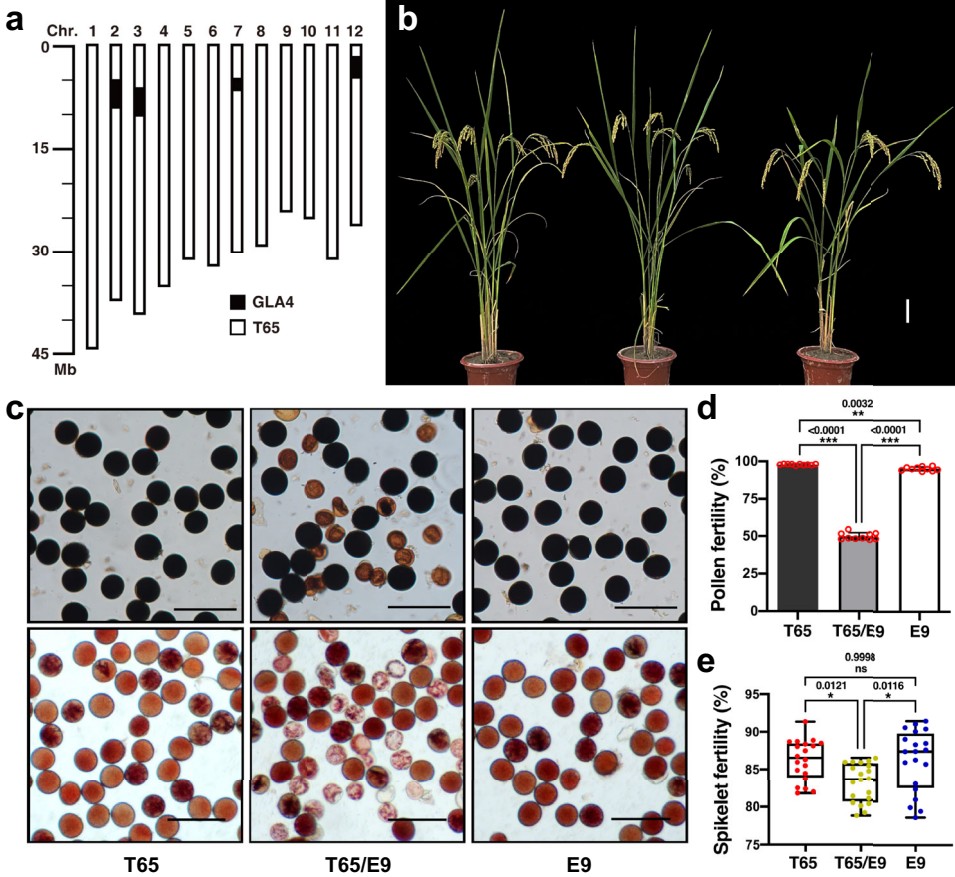

**Fig. 1 | HMS in T65/E9 $F_1$ hybrid. a** Location of substituted segments on the chromosomes of E9. Chromosomal segments from T65 (white boxes) and GLA4 (black boxes) are indicated. **b** Morphology of T65, T65/E9 $F_1$ hybrid and E9 plants. **c** Pollen stained with $I_2$-KI (upper panels) or TTC (lower panels) of T65, T65/E9 $F_1$ hybrids and E9. Scale bars = 100 μm. Pollen fertility (**d**) and

spikelet fertility (**e**) of T65, T65/E9 $F_1$ hybrids and E9, shown as means ± SD (*n* = 10 and 20 biologically independent samples for (**d**, **e**), respectively). Two-tailed Student's *t* tests were performed to determine significant differences (\**p* < 0.05, \*\**p* < 0.01 and \*\*\**p* < 0.001). Source data are provided as a Source Data file.

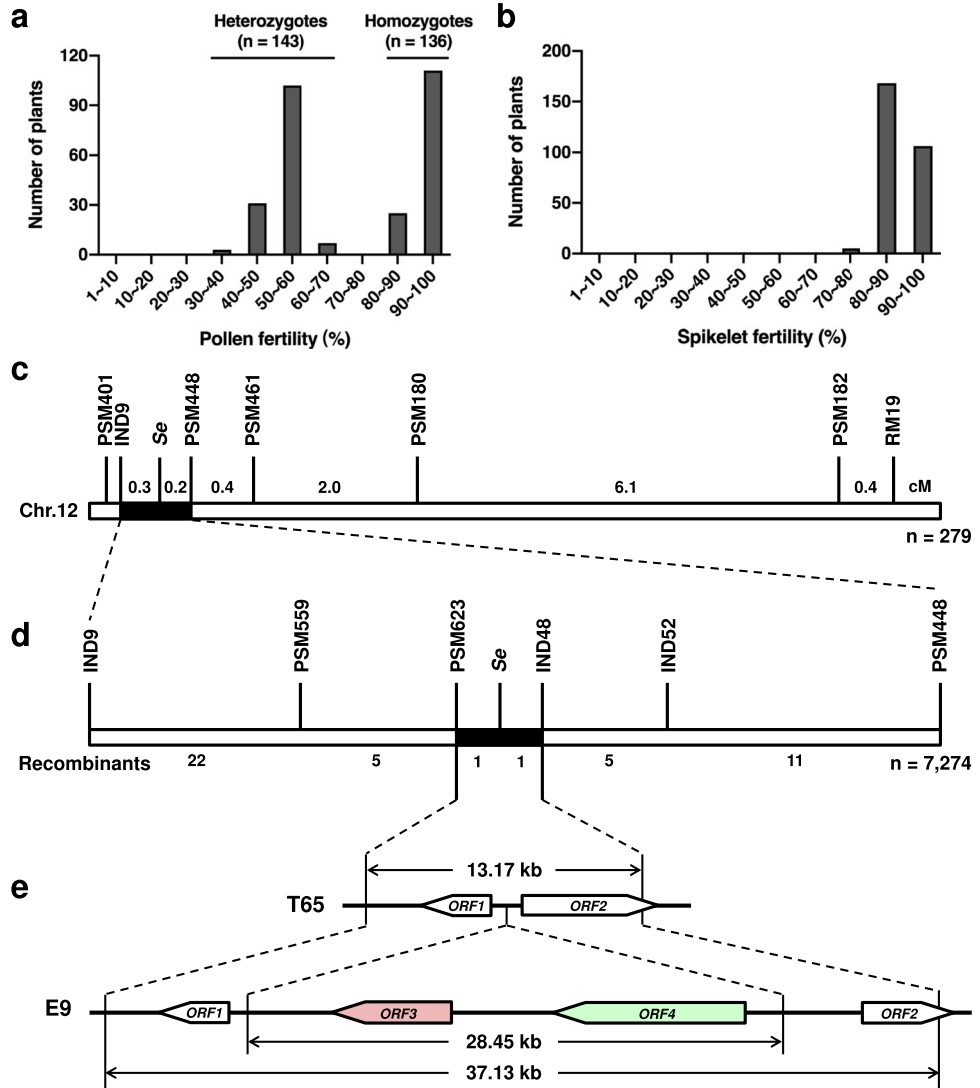

**Fig. 2 | Map-based cloning of *Se*.** Frequency distribution of pollen fertility (**a**) and spikelet fertility (**b**) in the T65/E9 F₃ segregating population. **c** *Se* was preliminarily mapped between IND9 and PSM448 on chromosome 12 using 279 T65/E9 F₃ plants. **d** *Se* was further delimited to a 13.17-kb region between PSM623 and IND48 using 7,274 T65/E9 F₄ plants. **e** *ORF*s predicted in the PSM623-IND48 interval. Arrows indicate the orientation of *ORF*s.

pollen fertility was restored to ~75% (Fig. 4a–c and Supplementary Table 7). By contrast, no effect on pollen fertility was observed in the transgene-negative plants (*T/E; -/-*) or F₁ plants with either the *ORF1*^E9 or *ORF2*^E9 transgene (Fig. 4b and Supplementary Table 7). Moreover, plants in an F₂ population derived from F₁ plants with one hemizygous *ORF4* transgene (*T/E; ORF4/-*) segregated as expected ($\chi^2 = 1.42 < \chi^2_{0.05,7} = 14.067$) (Fig. 4d, e and Supplementary Table 8). *Se*-heterozygotes with homozygous *ORF4* (*T/E; ORF4/ORF4*) were fully fertile, and F₂ plants with genotypes of (*T/T; ORF4/-*) and (*T/T; ORF4/ORF4*) were recovered (Fig. 4d, e and Supplementary Table 8). In contrast, no (*T/T*) plants could be obtained in segregating population derived from T65/E9 F₁ plants (Fig. 3c and Supplementary Table 9). These results indicated that *ORF4* rescues T65-type pollen in a gametophytic manner. This notion was further supported by the partial restoration of pollen fertility (~75%) in hemizygous T₀ *ORF4* over-expression transgenic plants with a T65/E9 genotype (Supplementary Fig. 15 and Supplementary Table 9). In addition, we constructed *orf4* mutants using the CRISPR/Cas9 genome editing system. Three hemizygous *orf4* mutants were obtained, and produced semi-sterile pollen, as expected (Supplementary Fig. 16 and Supplementary Table 10). Furthermore, no *orf4* homozygous mutant could be recovered in T₁ progeny of these hemizygous *orf4* plants (Supplementary Table 11). In

contrast, *orf4* homozygous mutants could be easily obtained when *ORF3* was also mutated. As expected, *orf3orf4* double mutants with T65/E9 genotypes showed normal pollen fertility (Supplementary Fig. 17).

Although both ORF3 and ORF4 localized in cytoplasmic foci (Supplementary Fig. 18), no direct interaction of ORF3 and ORF4 was detected using a yeast two-hybrid system (Supplementary Fig. 19). qRT-PCR analysis revealed that *ORF3* and *ORF4* are expressed broadly in different tissues and in all stages of anther development. The expression level in E9 was about twice as high as that in the T65/E9 F₁ hybrids (Supplementary Fig. 20), reflecting the different copy numbers (one versus two) in these backgrounds.

## Evolution of the *Se* locus

To trace the origin of *Se* locus, we BLAST searched ORF3 and ORF4 homologs in the Genebank database and found that full-length homologs of ORF3 can only be found in *Oryza* species, indicating it is a gene arose within the *Oryza* lineage. In contrast, homologs of ORF4 are widely distributed in *Poaceae* and they all share the conserved protein domain (Supplementary Fig. 21), suggesting its conserved function across grasses.

To trace the evolutionary history of *Se* locus in the *Oryza* genus, we first assayed the presence or absence of the locus by analyzing

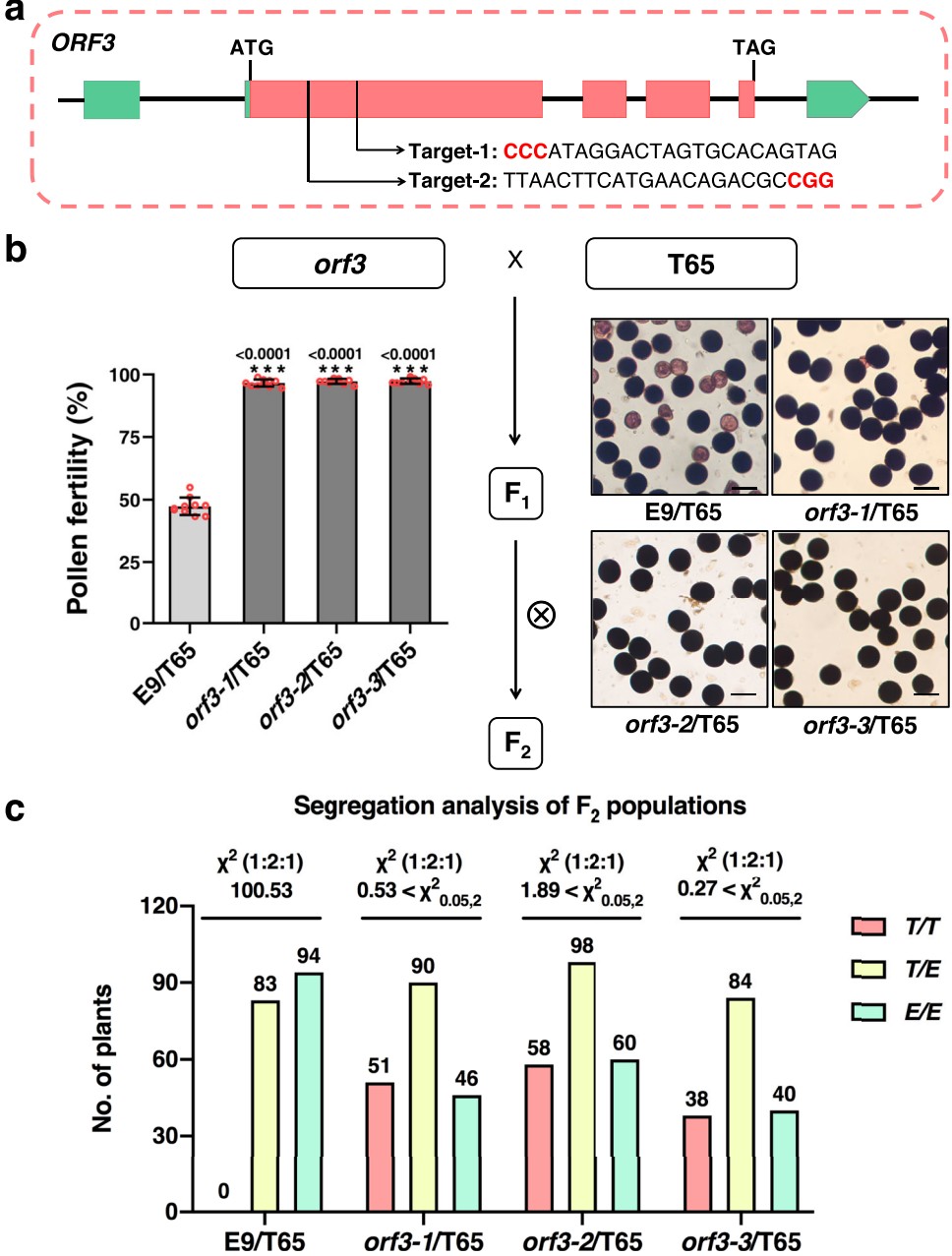

**Fig. 3 | Loss-of-function of *ORF3* eliminates *Se*-mediated HMS and segregation distortion. a** Position of two editing targets in *ORF3*. PAM sequence is highlighted in red. **b** F₁ hybrids with three *orf3* alleles exhibited normal pollen fertilities. Pollen fertility is shown as means ± SD (*n* = 10 biologically independent samples). Two-tailed Student's *t* tests were performed to compare the pollen fertility of *orf3*/T65 F₁ hybrids and E9/T65 F₁ hybrids (***p < 0.001). Scale bars = 50 μm. **c** Segregation of pollen fertility in F₂ progeny derived from E9/T65 or *orf3*/T65 crosses. *T/T*, *E/E* and *T/E* represent the homozygous genotype for T65 and E9 and the heterozygous genotype, respectively. Source data are provided as a Source Data file.

publicly available high-quality genome sequences of 59 *O. sativa*[4, 6,48] and 1 *O. glaberrima*[6], and PCR amplifying the region spanning part of *ORF3* and *ORF4* (Supplementary Fig. 22) in a large *Oryza* panel consisting of 787 individuals from 14 *Oryza* species. Consistent with the BLAST search analysis, target sequences were successfully identified from 363 out of 830 individuals of AA genome, but failed to be amplified from any individuals of GG (*O. meyeriana*, 3/3), FF (*O. brachyantha*, 2/2), EE (*O. australiensis*, 3/3), CC (*O. eichingeri* and *O. officinalis*, 5/5) and BB (*O. punctata*, 4/4) karyotypes (Fig. 5a and Supplementary Table 12). These results show that the *Se* locus is polymorphic in the AA rice species, but absent in *Oryza* species of other karyotypes, suggesting the emergence of the locus is specifically associated with the history of AA-genome *Oryza* species. Within AA

species, the locus can be found in Asian cultivated rice species complex, including *O. sativa* (230/384), *O. nivara* (10/44) and *O. rufipogon* (107/342) and African cultivated rice species complex including *O. glaberrima* (7/7) and *O. barthii* (4/4) (Fig. 5a and Supplementary Table 12). Surprisingly, it is also present in the basal species of AA genome, *O. meridionalis* (5/5), but absent in the rest species including *O. longistaminata* (28/28) and *O. glumaepatula* (16/16) (Fig. 5a and Supplementary Table 12). Further sequence analyses on ORF3 in individuals containing the *Se* locus identified 14 different haplotypes, among which 13 haplotypes confer truncated form of ORF3 protein, and the haplotype conferring full length protein can only be found in Asian cultivated rice species complex (Supplementary Data 1). We tested the effect of truncations by verifying the function of a common

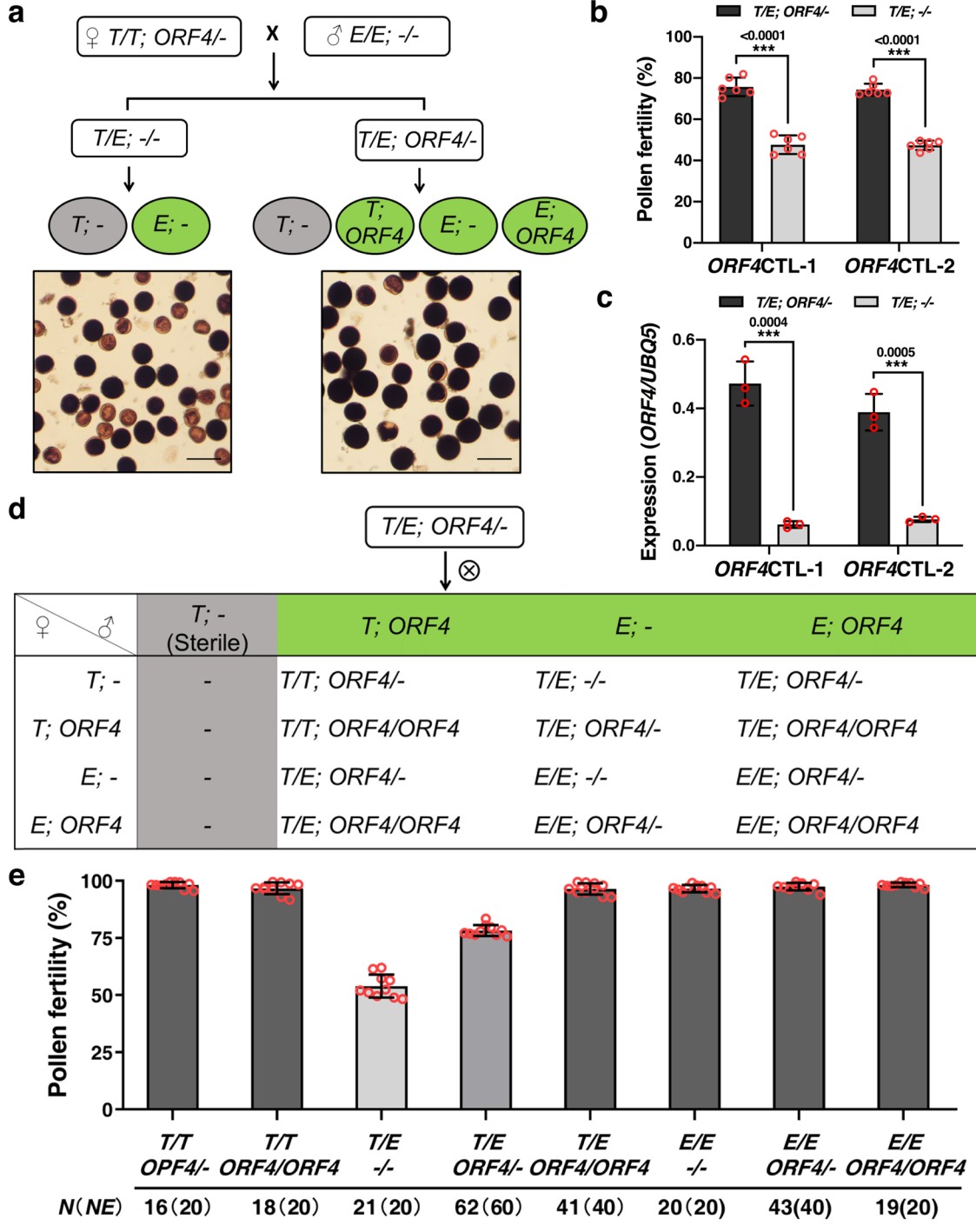

**Fig. 4 | *ORF4* rescues T65-type pollen in a gametophytic manner. a** *ORF4* hemizygous transgenic $T_0$ plants were crossed with E9 to generate $F_1$ plants without the *ORF4* transgene (*T/E; -/-*) (left, ~50% pollen fertility) and $F_1$ plants with the *ORF4* transgene (*T/E; ORF4/-*) (right, ~75% pollen fertility). Scale bars = 100 μm. **b** Comparison of pollen fertility between $F_1$ plants with (*T/E; ORF4/-*) and without (*T/E; -/-*) the *ORF4* transgene. Pollen fertility is shown as means ± SD ($n = 6$ biologically independent samples). **c** Comparison of *ORF4* expression between $F_1$ plants with (*T/E; ORF4/-*) and without (*T/E; -/-*) the *ORF4* transgene. Data are presented as means ± SD ($n = 3$ biologically independent experiments). **d** A schematic showing expected genotypes in $F_2$ progeny from selfing of $F_1$ plants with a single hemizygous copy of the *ORF4* transgene (*T/E; ORF4/-*). **e** Observed genotype frequencies and pollen fertility of $F_2$ progeny from a selfed $F_1$ plant (*T/E; ORF4/-*). The observed (*N*) and expected (*NE*) numbers of individual genotypes among 240 $F_2$ progeny are shown. Pollen fertility is shown as means ± SD ($n = 10$ biologically independent samples). Significant differences were determined by two-tailed Student's *t* tests (***$p < 0.001$). Source data are provided as a Source Data file.

truncation form (ORF3^Hap-2) and found that its subcellular localization was disrupted (Supplementary Fig. 18) and the toxicity effect was abolished in transgenic experiments in yeast and rice callus (Supplementary Figs. 13, 14). These results show that the truncation renders ORF3^Hap-2 non-functional and they also suggest that full-length sequence might be necessary for the ORF3 function, and if so, the

functional ORF3 is confined to the Asian cultivated rice species complex, with a significant enrichment of accessions from *O. sativa indica* subspecies. In contrast, truncated haplotype at ORF4 is extremely rare, which is only found in one *indica* individual out of 770 accessions assayed in the panel consisting of *O. sativa, O. nivara* and *O. rufipogon* (Supplementary Data 2). A 369-bp sequence presence/absence

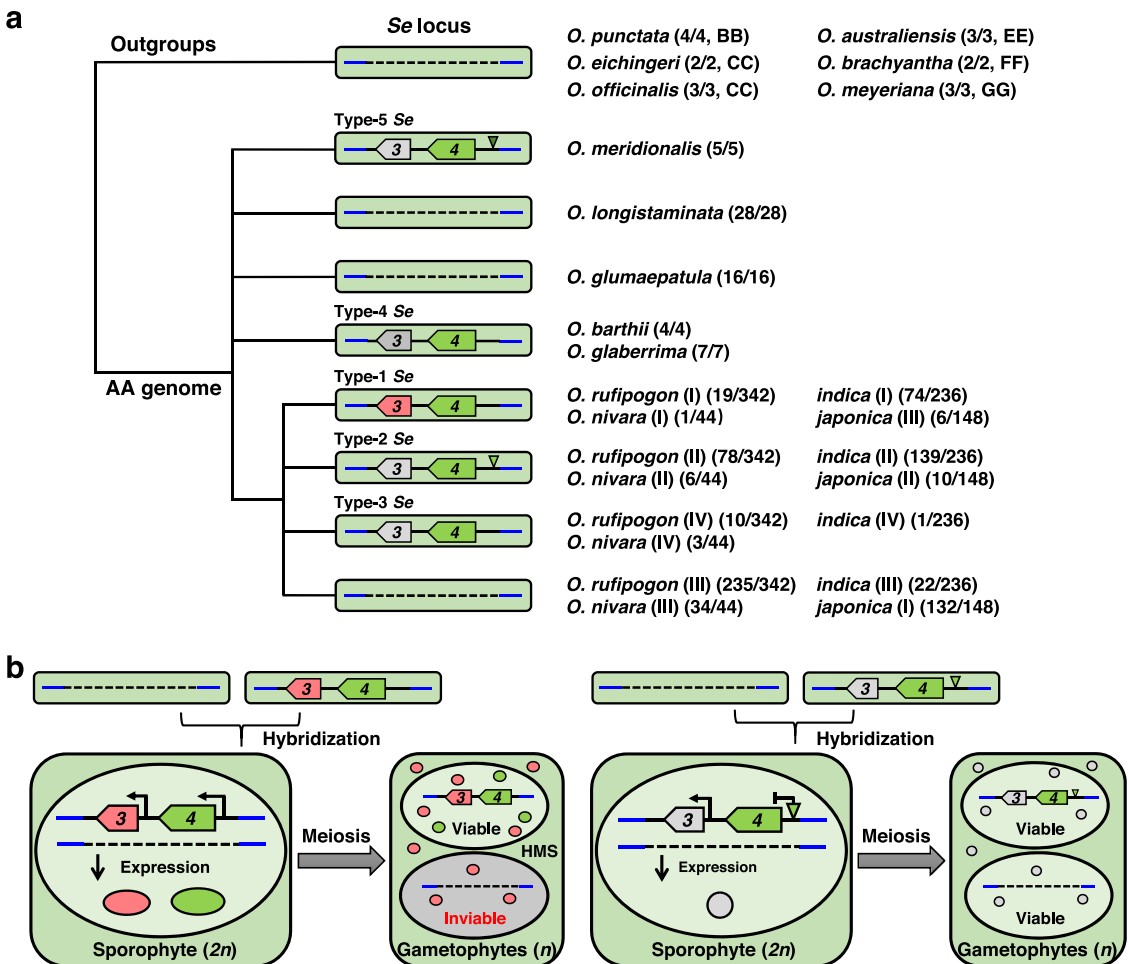

**Fig. 5 | Model for the evolutionary history and genetic action of *Se*. a** Proposal for the evolutionary history of *Se* locus. Ancestral wild rice lack the PAV region, and *Se* locus emerged after the divergence of the AA-genome clade of *Oryza*. Following the differentiation of *O. meridionalis*, *O. barthii*, and *O. rufipogon*, the *Se* locus later evolved in parallel but independent lineages. Based on the haplotype combinations of *ORF3* and *ORF4*, five main types of *Se* loci have been identified. *O. rufipogon*, *O. nivara*, *indica* and *japonica* samples include four, four, four, and three different types, respectively, and their frequencies are shown next to each type. Full-length *ORF3* (red), truncated *ORF3* (gray), *ORF4* (green), and deletions in *ORF4* promoter (triangles) are shown. **b** A proposed model for the molecular genetic basis of *Se*-mediated HMS. In hybrids with hemizygous functional *ORF3* and *ORF4* (left), pollen lacking the PAV region are aborted because of the pollen-killing effect of *ORF3* and lack of protective *ORF4*. When truncated and non-functional *ORF3* is present (right), only fertile pollen are produced.

variation was identified in the promoter of *ORF4* in the panel (Supplementary Data 2), and the deletion of the sequence did not affect the subcellular location of ORF4 (Supplementary Fig. 23) but significantly reduced the expression level (Supplementary Fig. 24), suggesting it is a functional variant in the rice population. A total of 37 haplotypes of the *Se* locus were identified when *ORF3* and *ORF4* were considered together, and could be further divided into five *Se* types according to the function and origins of *ORF3* and *ORF4* (Fig. 5a and Supplementary Data 3). Types 1 to 3 exist in Asian cultivated rice species complex of *O. sativa*, *O. nivara* and *O. rufipogon*, with Type-1 and Type-2 as the two major types representing 27.55% and 64.19% of the rice accessions with the PAV region, respectively (Fig. 5a and Supplementary Tables 13, 14), while Type-4 and Type-5 *Se* are haplotypes of African cultivated rice species complex and *O. meridionalis*, respectively (Fig.5a and Supplementary Tables 13, 14). In sum, our analyses suggest that *ORF3* is possibly formed in the Asian rice lineage, and its functional form is maintained in *O. sativa indica* subspecies in moderate frequency (74/236) (Fig. 5a and Supplementary Tables 13, 14). While in *japonica*, the majority varieties (132/148) have both *ORF3* and *ORF4* deleted (Fig. 5a and Supplementary Table 12). Therefore, the *Se* locus is an important isolation barrier between *indica* and *japonica*, as it is highly likely to cause hybrid incompatibility when individuals from the two subspecies interbreed.

## Discussion

PAVs represent a class of SVs that has been suggested to contribute to individual adaptation[49,50]. Although PAVs have been mostly linked to roles in ecological and agronomic adaptation in rice[4,7], little is known about possible involvement in speciation. In this study, we identified a PAV contributing to *indica*-*japonica* subspecies reproductive isolation through HS.

HS maintains the integrity of a species over time, reducing or directly restricting gene flow to or from other species[23]. Consistent with the complex nature of one-locus HS loci identified in previous studies[25–33], the *Se* locus in the PAV region studied here comprises two adjacent genes, *ORF3* and *ORF4*. In hybrids with functional *ORF3* and *ORF4*, pollen lacking the PAV is aborted. Such pollen lacks protective ORF4 to counteract the pollen-killing effect of ORF3 (Fig. 5b). Interestingly, *ORF3* expression in transgene-positive lines from two independent *ORF3* complementation and one overexpression experiments was less than 1/6 of that in T65/E9 F$_1$ hybrids (Supplementary Fig. 12), and no *orf4* homozygous CRISPR/Cas9 mutants were obtained in T$_0$ or T$_1$ generation (Supplementary Tables 10, 11). These indicate that *ORF3* is toxic to living cells without *ORF4*, and its expression should be strictly regulated. The toxicity of ORF3 is also supported by the growth inhibition in T65 callus transformed with *ORF3* (Supplementary Fig. 13)

and in yeast cells after ORF3 induction (Supplementary Fig. 14). However, one homozygous mutant containing a deletion in *pf12B* (allelic to *ORF4*) was identified under BL (*pf12*-NJ/NJ) genetic background[34]. One possibility is that this mutant *pf12B* allele might still preserve some weak function. Genetic verification of this mutant *pf12B* allele would clarify its functionality. Another possibility is that different copies of ORF4 homologs might exist in BL genetic background, and some ORF4 homologs might be functional, which also needs to be further studied.

Consistent with the predicted function of ORF3 as a superfamily II DNA and RNA helicase, ORF3 and ORF4 co-localized in cytoplasmic foci known as Processing bodies (P-bodies) and Stress Granules (SGs) (Supplementary Fig. 18b,e), which play critical role in mRNA metabolism, including processes such as RNA decay, storage and translation repression[51,52]. Plants deficient in P-body components display severe developmental perturbations such as postembryonic lethality and seedling lethality[53]. Correspondingly, transcriptomic analyses revealed that *pf12A* induced a drastically increased number of differentially expressed genes in the anthers at the mature pollen stage possibly due to aberrant base excision repair and RNA degradation in the gametes[34]. Although the exact role of ORF3 and ORF4 in cytoplasmic foci remained to be further studied, our results showed the ORF3 didn't affect the formation of P-bodies and SGs (Supplementary Fig. 25). In addition to localization in cytoplasmic foci, ORF3 also showed localization in mitochondria (Supplementary Fig. 18g), suggesting the abortion of pollen might be partially due to impaired mitochondria function and possibly poor energy supply, as shown in male sterilities caused by mitochondria-expressed genes, such as *orfH79*, *WA352*, *S27/S28* and *qHMS7-ORF2*[27,36,54,55]. The induction of functional ORF3 in INVSc1 yeast cells significantly inhibited the growth of the transformants, while this growth inhibition was attenuated when *ORF3*[Δ100], which encodes ORF3 without mitochondrion localization signal showing disrupted mitochondria localization (Supplementary Fig. 18h), was transformed (Supplementary Fig. 14). In addition, Hyg-resistant secondary calli were apparently propagated from *Agrobacterium* co-cultivated primary calli transformed with *ORF3*[Δ100] (Supplementary Fig. 13a, b). In contrast to low expression of the transgene in *ORF3* transgene-positive calli, high expression level of transgenes, although not as high as negative controls, could be obtained in *ORF3*[Δ100] transgene-positive calli (Supplementary Fig. 13c), suggesting that the mitochondria and cytoplasmic foci localization of ORF3 are both essential for its function.

The PAV is identified between different subspecies of *O. sativa* which is AA-genome. The outgroups BB-genome *O. punctata*, CC-genome *O. eichingeri*, CC-genome *O. officinalis*, EE-gnome *O. australiensis*, FF-genome *O. brachyantha* and GG-genome *O. meyeriana* lack the PAV region, indicating the absence of PAV is the ancestral condition of the *Se* locus in *Oryza* species (Fig. 5a). The PAV is only found in AA genome of *Oryza* species, and remains polymorphic across different species in Asian and African cultivated rice species complexes, surprisingly, it is also found in basal species of AA genome *O. meridionalis* but not in other AA species including *O. glumaepatula* and *O. longistaminata*. This pattern could have been caused by recent gene introgression at the *Se* locus across different species or independent loss of the *Se* locus in *O. glumaepatula* and *O. longistaminata*. These results clearly show that the locus was formed within the AA genome species of *Oryza* genus, and particularly, *ORF3* is only found within Asian cultivated rice species complex.

Monophyletic origin[56], polyphyletic origin[7,57], and *japonica*-to-*indica* introgression theories[58] have been advanced to explain the origin and domestication of Asian cultivated rice. We found 89.19% *japonica* (*japonica* (I)) lack the PAV and 90.68% *indica* (*indica* (I), *indica* (II) and *indica* (IV)) contain the PAV (Fig. 5a and Supplementary Table 12). Moreover, the PAV is also segregating in the population of *O. rufipogon*, the presumed wild progenitor of domesticated rice. These patterns are consistent with the scenario that *japonica* and *indica* originated from different *O. rufipogon* populations respectively and showed different haplotype distributions at the locus as a result. Therefore, the haplotype diversity pattern at the *Se* locus is consistent with polyphyletic origin and independent domestication of *indica* and *janopica* from different ancestral *O. rufipogon* populations.

In *indica* (II)/*japonica* (I) F[1] hybrids, no gametes were selectively aborted through *Se* function due to the lack of pollen killer *ORF3*, generating minor types such as *indica* (III) (9.32% *indica*) and *japonica* (II) (6.76% *japonica*) (Fig. 5). Through repetitive backcrosses of *indica* (I) with *japonica* (I) during artificial selection, the rare *japonica* (III) (4.05% *japonica*) could also be created (Fig. 5a). For example, pedigree analyses indicated that *japonica* (III) varieties WYG7 and LG31 were bred through repeatedly backcrossing *indica* varieties derived from IR54 or TDK with different *japonica* varieties (Supplementary Fig. 26). Our results demonstrate that the differentiation of two subspecies of *O. sativa* L. can be partially reversed through artificial selection, which is consistent with the genetic mixture of *indica* and *japonica* shown in the evolutionary study of *S5*[59].

Despite clear examples of PAV controlling agronomic traits in different species, PAV events have been proposed to be largely restricted to genes with signatures of reduced essentiality, so called dispensable genes, and are recent additions to plant genomes[60]. Our study presents a PAV contributing to reproductive isolation of the two subspecies of Asian cultivated rice. Although the origin and maintenance mechanisms of the PAV remain unclear for the moment, the near fixation of the PAV in *indica* suggests it might be favorable in environment adaptation and/or human selection during the *indica* domestication process. Future study of *ORF3* and *ORF4* in response to different environmental cues may help to elucidate its beneficial effects in adaptation.

Reproductive isolation hinders the utilization of strong hybrid vigor in *indica*-*japonica* crosses in rice. Besides major effect on HMS, *Se* was also found to affect female gamete development at a lower degree (Supplementary Table 2 and Fig. 1e), which is consistent with result of *pf12A*[34]. Therefore, *Se* is one of the most important target to overcome *indica*-*japonica* HS. In our study, Type-3 *Se* allele with non-functional *ORF3* and full-length *ORF4* was identified as a neutral haplotype in *O. rufipogon* (IV), *O. nivara* (IV) and *indica* (IV) (Fig. 5a and Supplementary Data 3), and could be crossed into current varieties to build natural hybrid-compatible lines. Alternatively, a neutral *Se* allele could be artificially generated by disrupting *ORF3* using CRISPR/Cas9 knockout or base editing strategies. Through genetic modification of *Se* and other *indica*-*japonica* hybrid sterility loci such as *SS*, *Sa* and *Sc*, widely-compatible lines could be created to break down reproductive barriers in *indica*-*japonica* hybrid rice breeding, thus having important implications in agriculture.

## Methods
### Plant materials and growth conditions
*O. sativa* ssp. *japonica* T65, NIL E9 and T65/E9 F[1] hybrid were used in gene cloning and functional analysis. The wild and cultivated rice species used for haplotype investigation and evolutionary analysis came from our laboratory or were provided by the International Rice Research Institute (Los Baños, Philippines), Guangxi Academy of Agricultural Sciences (Guangxi, China) and Yunnan Academy of Agricultural Sciences (Yunnan, China). All plant materials were grown at the experimental farm of South China Agricultural University, Guangzhou, China.

### Phenotypic characterization
To analyze pollen fertility, panicles at the flowering stage were fixed in FAA fixative [89% (v/v) ethanol, 6% (v/v) formaldehyde and 5% (v/v) acetic acid], and more than 200 pollen grains stained with 1% potassium iodide ($I_2$-KI) solution were observed with an optical microscope (BK6000, OPTEC, China) to obtain pollen phenotype. To assess pollen

vitality, pollen grains at the anthesis stage were collected and stained with 0.1% 2,3,5-triphenyl-2h-tetrazolium chloride (TTC) solution for 30 min before microscopic examination. A stereo microscope (SZ810, OPTEC, China) was used to observe the spikelets and anthers. Spikelet fertility was measured as the seed-setting rate of 20 individual main panicles.

### Cytological analysis

For scanning electron microscopy (SEM) and transmission electron microscopy (TEM) observations, mature anthers of T65, E9 and T65/E9 $F_1$ hybrid were pre-fixed in 0.1 M sodium phosphate buffer containing 4% glutaraldehyde (pH7.0) at 4 °C overnight. For SEM, pre-fixed samples were rinsed 3 times with 0.1 M phosphate buffer (pH7.0), and then were post-fixed for 2 h with 1.5% osmium tetroxide ($OsO_4$) in PBS (pH7.0). After rinsing with PBS (pH7.0) and ethanol dehydration (30% to 100%), samples were exchanged 3 times with isoamyl acetate. Then, the fixed samples were dried with a critical point drier and coated with gold. The anther and pollen grains were observed using a scanning electron microscope (EVO MA 15, ZEISS, Germany) with an acceleration voltage of 10 kV. For TEM, pre-fixed samples were rinsed 3 times with PBS (pH7.0), post-fixed with 1.5% $OsO_4$ in PBS (pH7.0) for 2 hours, and rinsed again 3 times with PBS (pH7.0). The post-fixed anthers were subjected to ethanol dehydration (30% to 100%) and transferred gradually to 100% resin. Super-thin sections (60 nm) were collected on uncoated nickel grids, and double stained with uranyl acetate and citrate aqueous solution as described previously[61]. Images were obtained with a transmission electron microscope (Talos L120C, FEI, America) operated at 80 kV.

To observe the development of anther and pollen, anthers at different stages were pre-fixed in FAA fixative, and were then embedded using Technovit Embedding Kits (Technovit 7100, Germany). Semi-thin sections (2 μm) were prepared using a microtome (RM2235, Leica, Germany), followed by staining with 0.5% toluidine blue. The stained sections were imaged with an optical microscope (BK6000, OPTEC, China).

### Fine mapping of *Se* locus and sequence analysis

A mapping population was generated from a cross between T65 and E9. Two hundred and seventy-nine plants from the $F_3$ population were used to analyze the linkage between pollen fertility and SSR markers for primary gene mapping. Additional SSR and InDel markers were designed in the IND9-PSM448 interval based on the polymorphisms between *japonica* Nipponbare and *indica* 9311. Fine mapping was performed using 7274 plants from the $F_4$ population, and the final region was further confirmed through genotypic segregation and phenotypic analysis of the progeny of key recombinants. The genomic DNA sequence of *Se* region was amplified from T65 and E9 using KOD-Plus-Neo DNA polymerase (TOYOBO, Japan) and assembled using SnapGene. All primers are listed in Supplementary Data 4.

### Full-length cDNA cloning and qRT-PCR analysis

Total RNA from young panicles of T65 and E9 were extracted using TRIZOL reagent (Invitrogen, America) according to the manufacturer's instructions. First-strand cDNA was reverse transcribed using a Hifair® III 1st Strand cDNA Synthesis Kit (YEASEN, China). The predicted coding regions of candidate genes were amplified from cDNA and sequenced. RACE assay was performed by nested PCR with the SMARTer RACE cDNA Amplification kit (Clontech, America) following the manufacturer's instructions. The anthers in different stages (meiocyte stage, microspore stage, early-bicellular stage, late-bicellular stage, and tricellular stage) from T65, E9 and T65/E9 $F_1$ hybrids, along with different tissues of E9 were selected for qRT-PCR assays according to a previous study[62], and the expression values were normalized to *UBQ5* transcript levels.

### Constructs for rice transformation

For complementation analysis, four genomic fragments (7,641-bp, 8,082-bp, 8,463-bp and 10,755-bp) were amplified from E9 and cloned into the binary vector pCAMBIA1300 to obtain the $ORF1^{E9}$CTL, $ORF2^{E9}$CTL, *ORF3*CTL, and *ORF4*CTL constructs. All constructs were introduced into T65 using *Agrobacterium*-mediated transformation. Transformation of *ORF3*CTL was conducted twice in independent experiments, and 34 transgene-positive lines were obtained in total.

Mutations of *Se* were constructed using the CRISPR/Cas9 vector system as previously reported[63], and target sites were designed by the web-based tool CRISPR-GE/targetDesign (http://skl.scau.edu.cn/)[64]. Constructs of $orf1^{T65}$, $orf2^{T65}$ and $orf1^{T65}orf2^{T65}$ were introduced into T65; $orf1^{E9}$ and *orf3* constructs were transformed into E9; and *orf3orf4* was introduced into T65/E9 $F_1$ hybrids. For generation of *orf4* mutants, constructs of three *orf4* with different target sites in the coding region of *ORF4* (target-1: CCGAAGATTACGAATCTTGGACA; target-2: CATATCTCTCACTTCGCGTATGG; target-3: TGTTCGCCGATGCGCCGTAACGG) were transformed into E9 or T65/E9 $F_1$ hybrids in four independent experiments (Supplementary Table 10), and two types of frameshift mutations in three independent hemizygous *orf4* mutants were obtained.

Because we didn't detect $ORF2^{E9}$ expression in the tested materials (Supplementary Fig. 8), we didn't construct CRISPR/Cas9 vector targeting $ORF2^{E9}$ to generate $orf2^{E9}$.

Overexpression constructs (*ORF3*OE and *ORF4*OE), in which coding sequence of E9 was driven by CaMV 35 S promoter, were transformed into T65 and T65/E9 $F_1$ hybrids, respectively, to verify the killer function of *ORF3* and the protector function of *ORF4*, respectively.

Rice transformations were performed by Edgene Biotechnology Co., Ltd. Wuhan, China.

For pollen sterility analysis and qRT-PCR analysis of single given $T_0$ transgenic plant, standard deviation was calculated from the 3 plants generated from a single positive callus. Three or more independent $T_0$ transgenic plants were used as biological replicates.

### *Agrobacterium*-mediated callus transformation

Overexpression constructs of *eGFP* (empty vector), *ORF3-eGFP*, $ORF3^{\Delta100}$-*eGFP* (with 5′ sequence encoding N-terminal 100-aa of ORF3 deleted), $ORF3^{HJX74}$-*eGFP* ($ORF3^{Hap-2}$-*eGFP*) were transferred into *Agrobacterium tumefaciens* strain EHA105. Sterilized mature seeds were inoculated on UINB medium and cultured at 28 °C in the dark. After 7 days of incubating, the calli were collected and used for transformation. After 3 days of cocultivation with *Agrobacterium* at 25 °C in constant darkness on N6D-Acetosyringone (AS) medium (30 g/L sucrose, 10 g/L glucose, 0.3 g/L casamino acids, 4.39 g/L N6D medium salt mixture, 2.878 g/L L-proline, and 2.8 mg/L 2,4-D, 20 mg/L AS, pH 5.4), the calli were washed two or three times with sterile water and rinsed once with sterile water containing 500 mg/L carbenicillin. Then, the calli were cultured on UINB medium containing 50 mg/L hygromycin B (Hyg, Merck), 400 mg/L carbenicillin for 30 days at 32 °C in constant light. Selection frequency of Hyg-resistant callus represents data from 3 groups of independent experiments, with ~50 transformed calli per group. Six (for *ORF3-eGFP*) and more than ten (for others) Hyg-resistant calli were used to test the expression of transgenes.

### Susceptibility assay with inducible *ORF3* expression

*S. cerevisiae* INVSc1 transformed with a vector harboring coding sequence of *ORF3* gene from E9 (pYES2-*ORF3* and pYES2-$ORF3^{\Delta100}$), HJX74 (pYES2-$ORF3^{HJX74}$) or empty vector (pYES2-CT) was pre-cultured overnight at 30 °C in SC-Ura liquid medium (Coolaber, China) containing 2% glucose (SCGlu-Ura medium). To induce the *ORF3* gene driven by *GAL1* promoter, the transformant was inoculated at an initial

concentration equivalent to OD600 = 0.1 into 50 mL of SC-Ura containing 3% galactose and 2% raffinose (SCGal-Ura medium), and cultured for 92 h at 30 °C. The growth of yeast cells was monitored by OD600. In the case of the reference experiment without induction of the *ORF3* gene expression, SCGal-Ura medium was replaced with SCGlu-Ura medium, and the culture time is 40 h.

## Subcellular localization of ORF3 and ORF4

For subcellular localization, the *ORF3* and *ORF4* coding region (minus the stop codon) were cloned into the pRTVcGFP vector, and plasmids were prepared using the FinePure EndoFree Plasmid Mini Kit (GEN-FINE, China). Constructs of *eGFP* (empty vector), *ORF3-eGFP*, *ORF3*^A100^-*eGFP*, *ORF3*^HJX74^-*eGFP*, *ORF4-eGFP* (*ORF4*^Hap-1^-*eGFP*), *ORF4*^Hap-2^-*eGFP*, *ORF4*^Hap-3^-*eGFP* and *ORF4*^Hap-12^-*eGFP* were transformed or co-transformed with plasmid encoding the cytoplasmic foci marker AtTZF1-mCherry[65] into rice protoplasts using the polyethylene glycol method. Following incubation for 14 to 20 h at 28 °C, fluorescence was observed using confocal microscopy (TCS SP8, Leica, Germany). Transformed protoplasts were stained with mitochondrion-specific dye MitoTracker Deep Red (Invitrogen, America) to view mitochondira signal.

## Phylogenetic and evolutionary analysis of *Se* locus

To identify putative homologs of ORF3 and ORF4, full-length protein sequences were used to query the NCBI database (https://www.ncbi.nlm.nih.gov/). Sequence alignment and neighbor-joining phylogenetic tree construction were performed using MEGA7 software (bootstrap replication = 1000).

To trace the evolutionary history of the *Se* locus, genomic DNA was extracted from 787 rice samples, including 4 *O. punctata*, 2 *O. eichingeri*, 3 *O. officinalis*, 3 *O. australiensis*, 2 *O. brachyantha*, 3 *O. meyeriana*, 28 *O. longistaminata*, 16 *O. glumaepatula*, 5 *O. meridionalis*, 4 *O. barthii*, 342 *O. rufipogon*, 44 *O. nivara*, 6 *O. glaberrima*, and 325 landraces (consisting of 217 *indica* and 108 *japonica*) from Rice Diversity Panel 2[66]. For all materials that contained the PAV region, *ORF3* and *ORF4* were sequenced and assembled using SnapGene. All of the sequences obtained and an additional 60 publicly available high-quality genome sequences[4,6,48] were used for haplotype investigation and evolutionary analysis. All rice materials are listed in Supplementary Data 5.

## Reporting summary

Further information on research design is available in the Nature Portfolio Reporting Summary linked to this article.

## Data availability

The CDS and amino acid sequences of *ORF1*^T65^ [https://www.ncbi.nlm.nih.gov/nuccore/OR253652], *ORF1*^E9^ [https://www.ncbi.nlm.nih.gov/nuccore/OR253653], *ORF2*^T65^ [https://www.ncbi.nlm.nih.gov/nuccore/OR253654], *ORF3* [https://www.ncbi.nlm.nih.gov/nuccore/OR253655] and *ORF4* [https://www.ncbi.nlm.nih.gov/nuccore/OR253656] are available at GenBank under accession numbers OR253652 to OR253656, as well as in Figshare [https://doi.org/10.6084/m9.figshare.23690625]. Since expression of *ORF2*^E9^ could not be detected (Supplementary Fig. 8), we consider it presence as putative and thus we didn't deposit its sequence to GenBank. *ORF2*^E9^ sequence used for alignment in Supplementary Fig. 7 was deduced from R498 reference genome annotation. Genealogical trees were deduced from the public available data on China Rice Data center (WYG7: [https://www.ricedata.cn/variety/varis/601512.htm]; LG31: [https://www.ricedata.cn/variety/varis/609156.htm]). To access the parental information of the variety, you can click the symbol "+" in the upper left corner of the description pages of WYG7 and LG31. The breeding history of the variety was inferred by continuously clicking the symbol "+" appearing next to the parental line. Source data are provided with this paper.

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

## Acknowledgements

We thank Xiangdong Liu (South China Agricultural University, China), Yuntao Liang (Guangxi Academy of Agricultural Science, China) and Cuifeng Tang (Yunnan Academy of Agricultural Science, China) for providing wild rice materials, thank Bihai Shi and Bin Hu (South China Agricultural University, China) for the assistance with confocal microscopic observation, thank Donna E. Fernandez (University of Wisconsin-Madison, USA) for reading and editing the manuscript, and thank Bo Wang and Wenjuan Hu (South China Agricultural University, China) for the help in the work related to *Agrobacterium*-mediated callus transformation. This

work was supported by National Natural Science Foundation of China (Grant No. 31571483 to Z.L.), Guangdong Provincial Natural Science Foundation (Grants No. 2019A1515012077, 2021A1515010446, and 2015A030313415 to Z.L.), Guangdong Provincial Key Laboratory of New Technology in Rice Breeding (Grant No. 2020B1212060047 to Z.L.), and start-up research grants from Science, Technology and Innovation Commission of Shenzhen Municipality to H.W.

## Author contributions

Z.L. conceived and designed the experiment. D.W. performed most of the experiments. H.W. analyzed the evolutionary data and wrote the section on the evolutionary history of *Se*. X.X. did the fine-mapping experiment. M.W. conducted experiments of genetic verification of *Se* function. Y.W., H.C., F.P., H.Z., Z.M., W.X., X.L., J.F. and M.Z. conducted some of the experiments. Z.F., T.Y., J.Z., B.L., Y.R., G.Z., C.L. participated in material development. Z.L. drafted the proposal and wrote the manuscript. All authors read and approved the final manuscript.

## Competing interests

Z.L., D.W., J.F., M.Z. and Z.M. are inventors on patent application (No. 202210609889.8) filed to the State Intellectual Property Office of China by South China Agricultural University that covers *ORF3* and *ORF4* sequences of the *Se* locus and their applications.
