## [Peer Review File · Nature Communications]

Two complementary genes in a presence-absence variation contribute to indica-japonica reproductive isolation in riceReviewers' Comments:

Reviewer #1:

Remarks to the Author:

This paper describes a unique PAV region in rice – a region that contains two genes ORF3 and ORF4 that appear to be associated with hybrid sterility. The ORF3 gene encodes a pollen killer, and fertility is rescued by the ORF4 gene. Interestingly, ORF3 acts in a sporophytic manner, while ORF4 in a gametophytic system.

This is a very interesting paper in that it identifies another set of hybrid sterility genes that may contribute to indica/japonica reproductive isolation. The authors are right in pointing out that this could have implications in crop improvement by facilitating indica/japonica hybridization. The genetic analysis and subsequent molecular and cell biological analysis looks competent and convincing.

The one issue I have is their statement in the abstract, Intro and Discussion that they think this PAV is a “key driver” of indica and japonica reproductive isolation. This is a strong statement but they do not have any evidence for it – that is, is it this locus and not other known HS genes in rice? What is their evidence that it is a “key driver”? I would reword to say that it “may contribute to differentiation” but more evolutionary analyses would be needed to elevate this gene as a key driver.

I also noticed several word choices in the writing that editorial assistance is needed. For example, they use the word “wildly” when they mean “widely”. Also, in at least two cases they describe the two genes as “ying and yang” – a better word would be to say they are “complementary”.

Reviewer #2:

Remarks to the Author:

In this manuscript, Wang et al. reported the identification of a hybrid sterility locus *Se* responsible for japonica-type pollen selective abortion in indica-japonica cross. Two tightly linked genes encoding another pollen killer-protector system to regulate hybrid sterility were found in a 28.40-kb insertion from indica variety GLA4 (or NIL E9). Then the authors analyzed DNA sequence differentiation among diverse rice germplasms including wild rice collections, to trace the evolutionary origin of the *Se* locus. This study expands our understanding of molecular mechanisms governing reproductive isolation. In addition, because *Se* is one of major hybrid sterility QTLs that hinder utilization of robust heterosis in hybrid rice breeding, cloning of *Se* should add new opportunities to test strategic designs for complete removal of the inter-specific hybrid sterility.

In all, this is an interesting study. However, concerns also raised and need to be addressed as listed below.

1. Specify the concept of “yin and yang” and how it interprets the work model at the *Se* locus (Lines 79-81). Cite original references if applicable.
2. L101 “... resulting in lower spikelet fertility of F1 hybrid...” Generally speaking, ~50% fertile rate of pollens from T65/E9 hybrid F1 is sufficient for pollination and plants should have normal seed setting. What caused the low spikelet fertility of the T65/E9 hybrid F1? Does this locus also regulate female gamete development? Please perform reciprocal test crosses to clarify contribution of male and female gametes to the claimed lower spikelet fertility.
3. A quite good number of marginal plants (e.g. with 30-40% or 60-70% pollen fertility) in Fig. 2a. Please sequence representative plants to see if there are other loci contributing to the marginal pollen fertility rates. Also, are the plants with 70-80% spikelet fertility in Fig. 2b those with 30-40% pollen fertility in Fig. 2a?
4. The authors performed the experiment of ORF3 CRISPR knockout showed that the ORF3 from E9 is a pollen killer. But neither the complementation nor overexpression of ORF3 could induce pollen abortion, owing to the several folds-low expression of ORF3 relative to its nature allele as shown by

expression analysis, or high level-expressing events failed to survive as hypothesized by the authors. Please use transgenic plants with ORF3 driven by an inducible promoter to verify this statement. In addition, orf2E9 plants were not created to study a possible role of ORF2E9 at Se. It is possible that there may be more than one member of gametes killer in the killer-protector system regulating rice hybrid sterility, such as S5 (Yang et al), S1(Xie et al.).

5. Knocking out of ORF4 is an important approach to demonstrate its role as a pollen protector. Unlucky, only one hemi orf4 out of 34 transgenic plants was obtained in E9 background. The authors are encouraged to try different target sites to produce more hemi plants. Data from multi-events will surely provide a stronger ground to support ORF4 as a protector. An important note for Supplementary Table 7: how comes the 98% pollen fertility in the 17 non-edited T65/E9 plants? Rather, ~50% pollen fertility is expected.

6. To explore the function of ORF3 and ORF4, subcellular localization of ORF3 and ORF4 was assessed using rice protoplast system. The authors stated that both ORF3 and ORF4 were localized in mitochondria. The mitochondrial signal of the ORF3 protein was annotated in supplementary figure 8. Please also annotate possible mitochondrial signal of ORF4. Please repeat subcellular localization assay for both ORF3 and ORF4 and provide high quality images. It is highly recommended to include in new assays the constructs with the annotated mitochondrial signals truncated and see if the mitochondrial localization is interrupted.

7. The picture of yeast-two-hybrid assay in supplementary figure 15 look like patchwork. The assay needs to be repeated to provide convincing result.

8. The manuscript is not clearly and succinctly written and needs to be edited carefully by a native English speaker. This is very IMPORTANT.

Reviewer #3:

Remarks to the Author:

Hybrid sterility is the principal mechanism of producing postzygotic reproductive isolation, which has great importance to speciation by preventing gene flow. Revealing the genetic basis underlying hybrid sterility can help us to understand deeply how to generating and maintaining species differentiation as well as conduct genetic improvement.

The authors of this manuscript found a locus Se containing two adjacent genes, ORF3 and ORF4, and the locus show a presence-absence variation (PAV) between two subspecies of rice, indica and japonica. Based on their experimental evidence, they believe that ORF3 encodes a protein to kill sporophytic pollens, whereas ORF4 can protect pollens from being killed in a gametophytic manner. Because japonica without the PAV region lacks the specific protection of ORF4, the pollens of japonica haplotype are aborted in F1 hybrid of the two subspecies, leading to the decrease of spikelet fertility. According to the evolutionary analysis of haplotype in Se locus, the authors think the PAV variation of two subspecies was inherited from their different wild progenitors respectively and the PAV drove the differentiation of two subspecies in the initial domestication stage.

There is no doubt at all that the authors devote a lot of time and energy to complete a original research paper, which will be of significance to the research on hybrid sterility, genetic incompatibility and speciation as well as genetic improvement. Though I believe that the manuscript is really worth publishing, there are still a few flaws in it.

The authors cited the phylogenetic research using chloroplast genomic data to show that *O. longistaminata* and *O. glumaepatula* are the most ancestral AA *Oryza* genome species. The two species and the outgroup GG-genome *O. meyeriana* have no PAV region, so the authors thought that the absence of ORF3 and ORF4 was the ancestral type of Se locus. However, according to the whole nuclear genomic data (Stein, J. C., et al. 2018. Genomes of 13 domesticated and wild rice relatives highlight genetic conservation, turnover and innovation across the genus *Oryza*. *Nature Genetics* 50(2): 285-296), the most ancestral AA *Oryza* genome species is *O. meridionalis*, in consisting with

previous researches using nuclear genes. BB-, CC-, EE- and FF-genome have closer relationship with AA genome than the current outgroup GG-genome. I also noticed that FF-genome own ORF3 and ORF4 gene(Supplementary Figure 17). Therefore, I suggest that the authors select suitable outgroup to conduct the analysis again.

The PAV variation also exists in rice wild progenitor species and author think the PAV also leading to reproductive isolation within the wild progenitor species(Fig.5). I think that they need to offer experimental evidences to support the idea.

The authors divided the history of rice cultivation into two stages, domestication and modern breeding. They also take two modern breeding varieties as example, illustrating that modern breeding has changed the inter-subspecies reproductive isolation, which generated in the early domestication stage. My question is why the authors did not exclude modern breeding varieties and only investigate landraces.

I am confused on the size of T65/E9 F2 population. It is 167 in segregation analysis and 279 in linkage analysis, but more than 500 in investigation of fertility (fig. 2a). The details about Identifying Se locus is quite insufficient. Only F2 and F3 population size as well as the information of markers are available. I suggest that the authors offer the relevant details in the supplementary file. In addition, though low express level can explain the failure of function complement experiment of ORF3, there is no direct evidence for the function yet.

Besides comments mention above, I have a few suggestions for the minor revisions.

Please check line 200. " in contrast to no (T/T) plants could be obtained in segregation population derived from (T/E) F1 plants (Supplementary Fig. 11)".

Please check Supplementary Figure 2. "(g-h)" should be "(g-i)".

Please check Supplementary Figure 22. "LG31 (a) and WYG7 (b)" should be "WYG7 (a) and LG31 (b)".

Please check Supplementary Table 1. "with a Student's t-test analysis" should be " with a chi square test analysis".

Dear reviewers,

Re: Manuscript ID: NCOMMS-22-22392A

Title: Two complementary genes in a presence and absence variation contribute to *indica-japonica* subspecies differentiation in rice

(Previous title: A presence and absence variation induces *indica-japonica* subspecies differentiation in rice)

Thank you for your precious comments and advice. Those comments are all valuable and very helpful for revising and improving our manuscript, and also have important guiding significance for our research. We have studied comments carefully and have made correction which we hope meet with approval. All amendments are highlighted with tracked changes in the revised version. In addition, point-by-point responses to the comments are listed below this letter.

We would love to thank you for allowing us to resubmit a revised copy of the manuscript and we highly appreciate your time and consideration.

Sincerely,

Ziqiang Liu

Response to reviewer's comments:

Reviewer #1:

This paper describes a unique PAV region in rice – a region that contains two genes *ORF3* and *ORF4* that appear to be associated with hybrid sterility. The *ORF3* gene encodes a pollen killer, and fertility is rescued by the *ORF4* gene. Interestingly, *ORF3* acts in a sporophytic manner, while *ORF4* in a gametophytic system.

This is a very interesting paper in that it identifies another set of hybrid sterility genes that may contribute to *indica/japonica* reproductive isolation. The authors are right in pointing out that this could have implications in crop improvement by facilitating *indica/japonica* hybridization. The genetic analysis and subsequent molecular and cell biological analysis looks competent and convincing.

Response: Thank you for your summary. We really appreciate your efforts in reviewing our manuscript.

The one issue I have is their statement in the abstract, Intro and Discussion that they think this PAV is a “key driver” of *indica* and *japonica* reproductive isolation. This is a strong statement but they do not have any evidence for it – that is, is it this locus and not other known HS genes in rice? What is their evidence that it is a “key driver”? I would reword to say that it “may contribute to differentiation” but more evolutionary analyses would be needed to elevate this gene as a key driver.

Response: We are grateful for your suggestion. We stated this PAV as a “key driver” of *indica* and *japonica* reproductive isolation based on the finding that 90.68% *indica* and 89.19% *japonica* showed the presence and

absence of the PAV region, respectively. The near fixation of the PAV in *indica* but near absence in *japonica* suggested the PAV played an important role in *indica-japonica* differentiation. Other known HS genes such as *S5*, *Sa* and *Sc* were also reported to be involved in the reproductive isolation between *indica* and *japonica*. Therefore, we thought all these HS genes were “key drivers” in *indica-japonica* differentiation process, and *Se* was one of them. We agree that it might be inappropriate to make such strong statement in our original manuscript. According to your advice, we have rephrased the PAV as a contributor to differentiation to avoid misunderstanding. The title was changed as “Two complementary genes in a presence and absence variation contribute to *indica-japonica* subspecies differentiation in rice”, and corrections have been made in the main text of the revised manuscript.

I also noticed several word choices in the writing that editorial assistance is needed. For example, they use the word “wildly” when they mean “widely”. Also, in at least two cases they describe the two genes as “ying and yang” – a better word would be to say they are “complementary”.

Response: Thank you for your careful review. We are very sorry for the mistakes in this manuscript and inconvenience they caused in your reading. For the two word choices you mentioned, we have made the corrections in Line 83, 89, 270 and 274. Furthermore, we have carefully revised the language issue with assistance of Dr. Donna E. Fernandez from University of Wisconsin-Madison, USA. We hope it can meet the journal’s standard.

Reviewer #2:

In this manuscript, Wang *et al.* reported the identification of a hybrid sterility locus *Se* responsible for *japonica*-type pollen selective abortion in *indica-japonica* cross. Two tightly linked genes encoding another pollen killer-protector system to regulate hybrid sterility were found in a 28.40-kb insertion from *indica* variety GLA4 (or NIL E9). Then the authors analyzed DNA sequence differentiation among diverse rice germplasms including wild rice collections, to trace the evolutionary origin of the *Se* locus. This study expands our understanding of molecular mechanisms governing reproductive isolation. In addition, because *Se* is one of major hybrid sterility QTLs that hinder utilization of robust heterosis in hybrid rice breeding, cloning of *Se* should add new opportunities to test strategic designs for complete removal of the inter-specific hybrid sterility.

In all, this is an interesting study. However, concerns also raised and need to be addressed as listed below.

Response: Thank you for your summary. We really appreciate your efforts in reviewing our manuscript. Our point-by-point responses are detailed below.

1. Specify the concept of “yin and yang” and how it interprets the work model at the *Se* locus (Lines 79-81). Cite original references if applicable.

Response: We adopted the concept of “yin and yang” from Chinese culture to describe two contradictory forces that come together to make a balanced and harmonious whole, which is similar to the PAV as a whole comprising two genes with opposite roles but combining in a complementary manner.

To avoid confusion, “The yin and yang effect” was rephrased as “The complementary effect” in Line 89 and 274 of the revised manuscript.

2. L101 “... resulting in lower spikelet fertility of F₁ hybrid...” Generally speaking, 50% fertile rate of pollens from T65/E9 hybrid F₁ is sufficient for pollination and plants should have normal seed setting. What caused the low spikelet fertility of the T65/E9 hybrid F₁? Does this locus also regulate female gamete development? Please perform reciprocal test crosses to clarify contribution of male and female gametes to the claimed lower spikelet fertility.

Response: Thank you for your careful review. In three-line hybrid rice breeding using gametophytic male-sterile lines such as BT-type and HL-type, F₁ hybrids produce semi-sterile pollen but have normal seed setting. Therefore, it is generally considered that 50% fertile pollen is sufficient for pollination and would not affect spikelet fertility. However, in hybrid rice production, spikelet fertility exceeding 80% is generally considered to be normal. In our study, spikelet fertility is 85.32±4.35% in T65/E9 F₁ hybrids, which is slightly lower than 91.51±3.65% in T65 and 92.42±3.41% in E9. Although statistical significance is observed, spikelet fertility of T65/E9 F₁ hybrid is still high enough to be considered “normal” in field production. For reciprocal test crosses, we presented data that test crosses with T65/E9 F₁ hybrids as the male parent and T65 or E9 as female parents generated only *T/E* or *E/E* progeny (Supplementary Table 2), indicating the male gametes derived from T65 were selectively aborted. According to your suggestion, we did test crosses with T65/E9 F₁ hybrids as the female parent and T65 or E9 as male parents in this October. Although we haven’t got the results yet, we believe that *Se* does not regulate female gamete development. Otherwise, spikelet fertility would be severely affected as shown in the cases of *S5* and *S7*, where spikelet fertility was much less than 85.32±4.35% of

T65/E9 F₁ hybrids (Yang *et al.*, 2012. A killer-protector system regulates both hybrid sterility and segregation distortion in rice. *Science* 337: 1336-1340; Yu *et al.*, 2016. Hybrid sterility in rice (*Oryza sativa* L.) involves the tetratricopeptide repeat domain containing protein. *Genetics* 203: 1439-1451). In addition, *Se* is located near and appears to be allelic to previous identified F₁ pollen sterility QTLs *S36/S25* and *S39(t)*, which were shown to only regulate male gamete development (Win *et al.*, 2009. Identification of two loci causing F₁ pollen sterility in inter- and intraspecific crosses of rice. *Breeding Science* 59(4): 411-418; Xu *et al.*, 2014. Mapping three new interspecific hybrid sterile loci between *Oryza sativa* and *O. glaberrima*. *Breeding Sciences* 63(5): 476-482). We have cited these articles in the Results section (Line 149) of our revised manuscript.

3. A quite good number of marginal plants (e.g. with 30-40% or 60-70% pollen fertility) in Fig. 2a. Please sequence representative plants to see if there are other loci contributing to the marginal pollen fertility rates. Also, are the plants with 70-80% spikelet fertility in Fig. 2b those with 30-40% pollen fertility in Fig. 2a?

Response: Thank you for your careful review. We got the data of original Fig. 2a-b more than ten years ago. The rice population used for fertility analysis is in the BC₆ generation, in which we could not rule out the possibility of minor effect QTLs. In addition, environment might also affect pollen and spikelet fertility, making it difficult to distinguish the effects between background QTLs and environment. As clear monogenic inheritance of pollen fertility was available at that time (original Fig. 2a), we focused on the analysis of the major effect gene *Se*.

After rechecking the data, there is some overlap between plants with 70-80% spikelet fertility in original Fig. 2b and those with 30-40% pollen fertility in original Fig. 2a. Unfortunately, we failed to get results of original Fig. 2a-b

before harvest, and missed to keep seeds of the marginal plants you mentioned. We apologize for not being able to sequence the representative marginal plants.

As Reviewer #3 raised confusion about the population size we used in the identification and fine-mapping of *Se*, we have replaced Fig. 2a-b with the data collected from 279 T65/E9 F₃ plants used in linkage analysis. Although there is still some marginal plants in this population, its number in the replaced Fig. 2a-b is much less than that in the original Fig. 2a-b.

4. The authors performed the experiment of *ORF3* CRISPR knockout showed that the *ORF3* from E9 is a pollen killer. But neither the complementation nor overexpression of *ORF3* could induce pollen abortion, owing to the several folds-low expression of *ORF3* relative to its nature allele as shown by expression analysis, or high level-expressing events failed to survive as hypothesized by the authors. Please use transgenic plants with *ORF3* driven by an inducible promoter to verify this statement. In addition, *orf2*^{E9} plants were not created to study a possible role of *ORF2*^{E9} at *Se*. It is possible that there may be more than one member of gametes killer in the killer-protector system regulating rice hybrid sterility, such as S5 (Yang et al), S1(Xie et al.).

Response: We deeply appreciate your comments and suggestion. Actually, we did complementation analysis of *ORF3* twice in independent experiments, and got 34 transgene-positive lines in total. We apologize for not presenting this information and have added it in the Methods sections (Lines 553-555) of our revised version. In all transgene-positive lines got from three independent transgenic experiments (two independent complementation and one overexpression analysis), *ORF3* showed very low expression. Combined with the results of the *ORF3* loss-of-function experiments, we think these

multi-events' data could justify our conclusion that *ORF3* has a killer function and its expression should be strictly regulated.

As you suggested, transgenic plants with *ORF3* driven by an inducible promoter would definitely not only better explain the pollen killer function of *ORF3*, but also provide excellent materials for studying the detailed molecular mechanism of *ORF3*. We are constructing plasmids for inducible expression of *ORF3*, as well as plasmids for RNAi of *ORF4*. However, if everything goes well, we would need half a year to get transgenic plants, and at least another half a year to do the analysis. We hope to publish these results, if valuable, in the future.

For *ORF2^{E9}*, we mentioned in the Methods section of our original manuscript that “No *orf2^{E9}* was designed and constructed, since no expression of *ORF2^{E9}* could be detected in all tested materials.” We designed 4 pairs of primers in *ORF2^{E9}*, but failed to detect its expression in different tissues and different stages (meiocyte, early-bicellular and tricellular stage) of anther development in E9 plants (Supplementary Fig. 8), suggesting the PAV in E9 disrupted the expression of *ORF2^{E9}*. Although *orf2^{E9}* plants were not created, we amplified a 8,463-bp E9 genomic fragment containing putative *ORF2^{E9}* and obtained *ORF2^{E9}CTL* plants in our complementation analysis. No effect on pollen fertility was observed in T65/E9 F₁ plants with *ORF2^{E9}CTL* transgene (Supplementary Table 6). Therefore, we think putative *ORF2^{E9}* is not involved in *Se*-mediated HMS.

Through map-based cloning of *Se*, recombinants R-6 and R-7 restricted the gametes killer to the 37.13-kb region in E9 genome (Fig. 2 and Supplementary Table 3). As all putative genes in this region from both T65 and E9 had been functionally analyzed through complementation and/or knockout analysis, we think *ORF3* is the only member of gamete killer.

5. Knocking out of *ORF4* is an important approach to demonstrate its role as a pollen protector. Unlucky, only one hemi *orf4* out of 34 transgenic

plants was obtained in E9 background. The authors are encouraged to try different target sites to produce more hemi plants. Data from multi-events will surely provide a stronger ground to support *ORF4* as a protector.

Response: Thank you for your advice. Actually, just as we repeated *ORF3* complementation analysis, we independently conducted *ORF4* knockout experiments twice. In the first knockout experiment, we selected two unique target sites in the coding region of *ORF4*, however, no *orf4* transgenic plant was obtained. Therefore, we did *ORF4* knockout experiment again by using only one target site, and got one hemizygous *orf4* mutant (*ORF4/orf4*) in the E9 background as we presented in our manuscript. We apologize for not presenting this information and have added it in the Methods section (Lines 561-565) of our revised version.

According to your advice, we are going to try different target sites to produce more hemizygous *orf4* mutants. Furthermore, we are constructing plasmids for RNAi of *ORF4*, with the purpose of getting transgenic plants with reduced expression of *ORF4* instead of knockout of *ORF4*. We hope to get interesting results worth publishing in the future.

An important note for Supplementary Table 7: how comes the 98% pollen fertility in the 17 non-edited T65/E9 plants? Rather, 50% pollen fertility is expected.

Response: We are extremely grateful to you for pointing out our mistake. It is actually $49.05 \pm 1.87\%$, as you expected. We have corrected this mistake in our revised manuscript.

6. To explore the function of *ORF3* and *ORF4*, subcellular localization of *ORF3* and *ORF4* was assessed using rice protoplast system. The authors stated that both *ORF3* and *ORF4* were localized in mitochondria. The

mitochondrial signal of the ORF3 protein was annotated in supplementary figure 8. Please also annotate possible mitochondrial signal of ORF4. Please repeat subcellular localization assay for both ORF3 and ORF4 and provide high quality images. It is highly recommended to include in new assays the constructs with the annotated mitochondrial signals truncated and see if the mitochondrial localization is interrupted.

Response: Thank you for the suggestion. We have repeated subcellular localization assay, and have updated images in Supplementary Fig. 15. Full-length ORF3 shows localization in both cytoplasmic foci and mitochondria, while deletion of mitochondrial targeting signal in ORF3^{Δ100} affects its mitochondrial localization but not cytoplasmic foci localization. No mitochondrial targeting signal was predicted in ORF4 by web-based tools including iPSORT (<https://ipsort.hgc.jp/index.html>) and MITOPROT (<https://ihg.helmholtz-muenchen.de/ihg/mitoprot.html>), and ORF4-GFP is targeted to cytoplasmic foci but not mitochondria.

Although it is still obscure about the molecular mechanism of *Se*, co-localization of ORF3 and ORF4 in cytoplasmic foci known as Processing bodies (P-bodies) and Stress Granules (SGs) is in consistent with the predicted function of ORF3 as a superfamily II DNA and RNA helicase. Plants deficient in p-body components display severe developmental perturbations such as postembryonic lethality and seedling lethality (Xu and Chua, 2011. Processing bodies and plant development. *Current Opinion in Plant Biology*, 14: 88-93), which may be why neither *ORF3* transformants with normal expression nor *orf4* homozygotes were obtained in our study. Accordingly, we have revised our description in the Results (Line 255) and Discussion (Lines 336-347) sections of our manuscript.

7. The picture of yeast-two-hybrid assay in supplementary figure 15 look like patchwork. The assay needs to be repeated to provide convincing result.

Response: Thank you for the suggestion. We have repeated the yeast-two-hybrid assay, and have updated images in Supplementary Fig. 16.

8. The manuscript is not clearly and succinctly written and needs to be edited carefully by a native English speaker. This is very IMPORTANT.

Response: We apologize for the language problems in the original manuscript. The language presentation was improved with assistance of Dr. Donna E. Fernandez from University of Wisconsin-Madison, USA. We hope it can meet the journal's standard.

Reviewer #3:

Hybrid sterility is the principal mechanism of producing postzygotic reproductive isolation, which has great importance to speciation by preventing gene flow. Revealing the genetic basis underlying hybrid sterility can help us to understand deeply how to generating and maintaining species differentiation as well as conduct genetic improvement.

The authors of this manuscript found a locus *Se* containing two adjacent genes, *ORF3* and *ORF4*, and the locus show a presence-absence variation (PAV) between two subspecies of rice, *indica* and *japonica*. Based on their experimental evidence, they believe that *ORF3* encodes a protein to kill sporophytic pollens, whereas *ORF4* can protect pollens from being killed in a gametophytic manner. Because *japonica* without the PAV region lacks the specific protection of *ORF4*, the pollens of *japonica* haplotype are aborted in F₁ hybrid of the two subspecies, leading to the decrease of spikelet fertility. According to the evolutionary analysis of haplotype in *Se* locus, the authors think the PAV variation of two subspecies was inherited from their different wild progenitors respectively and the PAV drove the differentiation of two subspecies in the initial domestication stage.

There is no doubt at all that the authors devote a lot of time and energy to complete a original research paper, which will be of significance to the research on hybrid sterility, genetic incompatibility and speciation as well as genetic improvement. Though I believe that the manuscript is really worth publishing, there are still a few flaws in it.

Response: Thank you for your summary. We really appreciate your efforts in reviewing our manuscript. Our point-by-point responses are detailed below.

The authors cited the phylogenetic research using chloroplast genomic data to show that *O. longistaminata* and *O. glumaepatula* are the most ancestral AA *Oryza* genome species. The two species and the outgroup GG-genome *O. meyeriana* have no PAV region, so the authors thought that the absence of *ORF3* and *ORF4* was the ancestral type of *Se* locus. However, according to the whole nuclear genomic data (Stein, J. C., et al. 2018. Genomes of 13 domesticated and wild rice relatives highlight genetic conservation, turnover and innovation across the genus *Oryza*. Nature Genetics 50(2): 285-296), the most ancestral AA *Oryza* genome species is *O. meridionalis*, in consisting with previous researches using nuclear genes. BB-, CC-, EE- and FF-genome have closer relationship with AA genome than the current outgroup GG-genome. I also noticed that FF-genome own *ORF3* and *ORF4* gene (Supplementary Figure 17). Therefore, I suggest that the authors select suitable outgroup to conduct the analysis again.

Response: Thank you for your careful review and suggestion. According to your advice, we analyzed the PAV in two samples each from the outgroups BB-genome *O. punctata* and CC-genome *O. eichingeri*, and found no PAV region. However, both putative *ORF3* and *ORF4*, named as *ORF3L* and *ORF4L*, could be separately amplified from different locations in BB- and CC-genome *Oryza*. Relatively low similarity was found between *ORF3* and *ORF3L*, as well as between *ORF4* and *ORF4L* (Supplementary Figs. 23 and 24). When the reference sequence of BB-genome *O. punctata* (Stein *et al.*, 2018. Genomes of 13 domesticated and wild rice relatives highlight genetic conservation, turnover and innovation across the genus *Oryza*. Nature Genetics 50: 285-296) was analyzed, *ORF3L* was found to be located approximately 36 kb upstream of *ORF4L* on chromosome 11 (Fig. 5a). In contrast, in AA-genome *Oryza*, *ORF4* is located next to *ORF3* promoter on chromosome 12. The location, organization, orientation and sequences of *ORF3L* and *ORF4L* suggest the absence of PAV is the ancestral condition of

the *Se* locus, and the PAV might have emerged from the *ORF3L-ORF4L* chromosome segment through a combination of recombination or translocation, deletion of intervening sequences and nucleotide mutations.

As *O. meridionalis* diverged earlier than *O. glumaepatula*, *O. barthii* and *O. rufipogon* (Stein *et al.*, 2018. *Nature Genetics* 50: 285-296; Gao *et al.*, 2019. *Evolution of Oryza chloroplast genomes promoted adaptation to diverse ecological habitats. Communications Biology* 2: 278), and the PAV region was present in all *O. meridionalis* tested, we propose that the PAV emerged in Gondwanaland before the differentiation of *O. meridionalis*, and was later lost in *O. glumaepatula* and in some *O. rufipogon*. After geographical separation, the existing PAV in *O. meridionalis*, *O. barthii* and *O. rufipogon* evolved in parallel but in independent lineages.

We have revised our description in the Results (Lines 285-286), Discussion (Lines 349-374) and Methods (Lines 597-598) sections, as well as in Fig. 5, Supplementary Figs. 23 and 24, and Supplementary Tables 9 and 13.

The PAV variation also exists in rice wild progenitor species and we think the PAV also leading to reproductive isolation within the wild progenitor species (Fig.5). I think that they need to offer experimental evidences to support the idea.

Response: We are grateful for your suggestion. We made the hypothesis of PAV leading to reproductive isolation within wild progenitor species based on the following findings. (1) The majority of *indica* (90.68%) and *japonica* (89.19%) showed the presence and absence of PAV, respectively; (2) *indica* accessions with PAV could be further divided into two major types, *indica* (I) with Type-1 *Se* and *indica* (II) with Type-2 *Se*, both existing with high frequency. However, if *Se*-mediated reproductive isolation happened within the progenitor rice species after the emergence of Type-2 *Se*, we would expect a larger proportion of *japonica* with Type-2 *Se* (*japonica* (II)) and

indica without PAV (*indica* (III)), since no reproductive isolation exists between *O. rufipogon* (II) and *O. rufipogon* (III), or between *indica* (II) and *japonica* (I) (Fig. 5b). Instead, only 6.76% *japonica* accessions and 9.32% *indica* accessions were found to contain Type-2 *Se* and have no PAV, respectively. Therefore, Type-1 *Se* likely arose first in ancestral *O. rufipogon* with PAV. The functional *ORF3* and *ORF4* in ancestral *O. rufipogon* would induce reproductive isolation with *O. rufipogon* (III), and the ancestral group would have separately domesticated to *indica*, which typically (214 out of 236) contain the PAV. In this lineage, a frame-shift mutation in *ORF3* and deletion in *ORF4* promoter subsequently occurred, generating Type-2 *Se*, and ancestral *O. rufipogon* was further divided into two subgroups, *O. rufipogon* (I) and *O. rufipogon* (II), which were domesticated to *indica* (I) and *indica* (II), respectively (Fig. 5a).

As also pointed out by Reviewer #1, we may have made too strong a statement about the evolutionary function of the PAV. To avoid misunderstanding, we have rephrased the PAV as a contributor to differentiation, and have changed the title as “Two complementary genes in a presence and absence variation contribute to *indica-japonica* subspecies differentiation in rice”.

The authors divided the history of rice cultivation into two stages, domestication and modern breeding. They also take two modern breeding varieties as example, illustrating that modern breeding has changed the inter-subspecies reproductive isolation, which generated in the early domestication stage. My question is why the authors did not exclude modern breeding varieties and only investigate landraces.

Response: Thank you for your careful review. We apologize for not describing our materials used in evolutionary analysis clearer. We sequenced *Se* locus of 217 *indica* and 108 *japonica* accessions belonging to Rice

Diversity Panel 2 (RDP2) which represents a world-wide diversity of rice. According to the literature describing RDP2 (McCouch *et al.*, 2016. Open access resources for genome-wide association mapping in rice. Nature Communications, 7: 10532), these accessions are landraces instead of modern varieties. As *Se*-mediated HS eliminates all male gametes derived from *japonica* in typical *indica/japonica* crosses, it would be extremely difficult for *japonica* with functional *ORF3* (*japonica* (III)) to be generated. To understand how rare *japonica* (III) was created, we analyzed publicly available high-quality genome sequences of 19 modern *indica* varieties and 40 modern *japonica* varieties, and introduced genealogical trees of two modern *japonica* varieties belonging to *japonica* (III). Since most accessions of our *O. sativa* are landraces, we feel “Artificial selection” might be a better word to replace “Modern breeding” in Fig. 5a. We have revised the relevant parts in the Discussion (Lines 426-436) and Methods (Lines 600-601) sections, as well as in Fig. 5a.

I am confused on the size of T65/E9 F₂ population. It is 167 in segregation analysis and 279 in linkage analysis, but more than 500 in investigation of fertility (fig. 2a). The details about identifying *Se* locus is quite insufficient. Only F₂ and F₃ population size as well as the information of markers are available. I suggest that the authors offer the relevant details in the supplementary file.

Response: We apologize for confusion in our manuscript about the identification of *Se*. Actually, we wasted a lot of time in the fine-mapping experiment, and several graduate students participated in this part of work in succession. At first, during the development of NILs including E9 in T65 background, we only checked the segregation of SSR markers in the T65/GLA4 BC₅F₂ generation and did not examine phenotype. Since segregation distortions of two SSR markers (RM19 and RM453) were found,

another student focused on the phenotype analysis, and determined monogenic inheritance of pollen sterility through analysis of the population with more than 500 individuals. However, she didn't keep leaves of these plants. Later, when we decided to map *Se*, we had to redo the linkage analysis again, and used 279 plants this time. Therefore, 167, 548 and 279 plants were in the T65/GLA4 BC₅F₂ generation, T65/E9 F₂ population, and T65/E9 F₃ population, respectively. To avoid misunderstanding, we have replaced Fig. 2a-b with the data collected from 279 plants used in linkage analysis. We are very sorry for the mistakes in our original manuscript, and have amended the relevant parts in the Results and Methods sections, as well as in Supplementary Fig. 1. In addition, details of the recombinants in the fine-mapping of *Se* were added in Supplementary Table 3 in our revised version.

In addition, though low express level can explain the failure of function complement experiment of *ORF3*, there is no direct evidence for the function yet.

Response: Thank you for your careful review and comments. Actually, we did *ORF3* complementation analysis twice in independent experiments, and got 34 transgene-positive lines in total. We apologize for missing this information in our original manuscript, and have added it in the Methods section (Lines 553-555) of our revised version. In all transgene-positive lines got from three independent transgenic experiments (two complementation and one overexpression analysis), *ORF3* showed very low expression. Combined with the results of the *ORF3* loss-of-function experiments, we think these multi-events' data could justify our conclusion that *ORF3* has a killer function and its expression should be strictly regulated.

As also suggested by Reviewer #2, we are constructing plasmids for inducible expression of *ORF3* and RNAi of *ORF4* to get materials for better

explaining the pollen killer function of *ORF3*. We still need time to get results, and hope to be able to share these results in another story in the near future.

Besides comments mention above, I have a few suggestions for the minor revisions.

Please check line 200. " in contrast to no (T/T) plants could be obtained in segregation population derived from (T/E) F₁ plants (Supplementary Fig. 11)".

Please check Supplementary Figure 2. "(g-h)" should be "(g-i)".

Please check Supplementary Figure 22. "LG31 (a) and WYG7 (b)" should be "WYG7 (a) and LG31 (b)".

Please check Supplementary Table 1. "with a Student's t-test analysis" should be "with a chi square test analysis".

Response: Thank you for your careful review and these corrections. We are sorry for these mistakes in our original manuscript, and have amended the these mistakes in our revised version.

Reviewers' Comments:

Reviewer #1:

Remarks to the Author:

For my part, the authors appear to have satisfactorily addressed my concerns.

Reviewer #2:

Remarks to the Author:

This is my second review of this paper by Dr Liu and colleagues. I am happy to see a great improvement in language with the assistance of Dr. Donna E. Fernandez at University of Wisconsin-Madison, USA. Majority of my comments have been addressed. While the conclusion on genetics of the killer-protector remains unchanged in large, there are new claims/interpretations on results (some of them generated from new experiments) in the revised version. Associated with those, I still have some comments/suggestions that need to be addressed to put the paper well-grounded on data.

1. The authors introduced a new claim on localization of ORF3 and ORF4 (P-bodies and SGs) to explore possible action mechanism of ORF3-ORF4, particularly of ORF3. Based on the predicted function of ORF3 as a superfamily II DNA and RNA helicase and the critical role of P-bodies and SGs in RNA metabolism (as referred by authors their deficiency causes postembryonic lethality), does the authors infer that ORF3 destroys P-bodies and SGs (or at least has negative impact on their formation)? I agree this is an interesting insight into ORF3's role as a killer. To support this, I believe it is important to repeat sub-cellular localization of at least ORF3, given the poor (inconvincible) images presented in Supplementary Fig. 16 and different results in the original manuscript. Also, pay attention to any changes of P-bodies and SGs when expressing ORF3 in protoplasts. I recommend using callus or suspension cells to prepare quality protoplasts. The authors did not touch how ORF4 detoxifies ORF3 (they are co-localized but do not interact directly), but I understand that it is a challenging job and could be pursued in future studies.

2. It seems that the authors down-value mitochondrion localization of ORF3 in the revised version. Considering the mitochondrion signal in ORF3 (with localization verified by independent protoplast assays) and the observation that ORF3 inhibits cell growth in yeast, however, the possibility that ORF3 functions in mitochondrion cannot be simply excluded. This concern can be addressed easily by doing complementary transformation with a signal-less version of ORF3. It takes about two months to complete the experiment since transgenic callus (not necessarily transgenic plants) can be used for analysis (to see if high level expression of mutated ORF3 in T65 can be obtained). Alternatively, the base editor system can be used to mutate the signal sequence in the T65/E9 F1 background to see pollen fertility in mutated plants. This BE approach will take longer time. Either experiment will clarify if mitochondrion localization is essential for functioning of ORF3.

3. I agree with Review #3 who gave constructive comments on analysis of evolutionary trajectory of ORF3-ORF4. Fig. 5a is confusing to me and I also don't see a clear description (I am not expecting conclusion) on evolution of ORF3-ORF4. Surely, the authors made appreciable efforts by analyzing numerous collections of both wild and cultivated rice. Evolution biology is not my field, so I am sorry I cannot give comments for improvement. The authors assume functioning of HAPORF4-2 in Type-1 Se, HAPORF4-22 to 25 in Type-3 Se and HAPORF4-30 to 34 in Type-4 Se, regardless of polymorphisms between them and lacking genetic verification. The authors should do genetic complementation in T65/E9 F1 background for 2-3 ORF4 haplotypes, particularly HAPORF4-2 in functional Type-1 Se, to verify their assumption.

4. Include controls, such as ORF3-AD+BD and ORF4-AD+BD, in Sup. Fig. 17.

Other minor points

1. Slightly lower spikelet fertility but statistically significant in T65/E9 F1 (Fig. 1d), which is consistent with the segregation ratio of T/E and E/E (54:74) from a cross with T/E as female in Supp. Table 2 (although not statistically significant). Those results imply that female gamete carrying T65 allele is slightly affected, but at a negligible degree relative to male gamete. Soften the claim rather than

simple argument.

2. For expression analysis (such as three ORF3 overexpression, two ORF4 overexpression transgenic T0 plants) and pollen fertility of single T0 plants (more examples throughout the paper), there is no information on biological replicates in either Method or figure legends. The authors need to indicate how SD was obtained from analysis of a single given plant.

3. Lines 143, change sterility to semi-sterility.

4. Lines 180-182, rephrase the sentence as Fig. 3a shows targets only for ORF3.

5. Supp Table 54 (also Line 204 in Text): typo of Supp Table 5?

6. Line 259, QRT-PCR. qRT-PCR is common.

7. Lines 430-433, soften the sentence (regarding full fertile pollen) as there are other genes controlling indica/japonica HMS.

8. Supp. Fig. 5w, the label MP could be BP?

9. Supp. Fig. 13, any copy number information of transgene? It seems the pollen fertility and segregation analysis were single-copy based.

10. Supp. Fig. 26 provides a good example on introgression of the PAV from indica to japonica. Give interpretation on how "artificial selection" was placed during the breeding procedure, at least in the legend.

Dear reviewers,

Re: Manuscript ID: NCOMMS-22-22392C

Current title: Two complementary genes in a presence and absence variation contribute to *indica-japonica* reproductive isolation in rice

(Previous title: Two complementary genes in a presence and absence variation contribute to *indica-japonica* subspecies differentiation in rice)

Thank you for your precious comments and advice. Those comments are all valuable and very helpful for revising and improving our manuscript, and also have important guiding significance for our research. We have studied comments carefully, have done additional experiments to address the raised questions, and have made corrections accordingly. Besides, we invited evolutionary biologist Hongru Wang (Agricultural Genomics Institute at Shenzhen, Chinese Academy of Agricultural Sciences) to evaluate our evolutionary data and have heavily revised the section on the evolutionary history of *Se*. All amendments are highlighted with tracked changes in the revised version, and our point-by-point responses to the comments are listed below this letter.

We highly appreciate your efforts in reviewing our manuscript.

\

Response to reviewer's comments:

Reviewer #1:

For my part, the authors appear to have satisfactorily addressed my concerns.

Response: Thank you for your time. We really appreciate your efforts in reviewing our manuscript.

Reviewer #2:

This is my second review of this paper by Dr Liu and colleagues. I am happy to see a great improvement in language with the assistance of Dr. Donna E. Fernandez at University of Wisconsin-Madison, USA. Majority of my comments have been addressed. While the conclusion on genetics of the killer-protector remains unchanged in large, there are new claims/interpretations on results (some of them generated from new experiments) in the revised version. Associated with those, I still have some comments/suggestions that need to be addressed to put the paper well-grounded on data.

Response: Thank you for your evaluation and comments. We really appreciate your efforts in reviewing our manuscript. Our point-by-point responses are detailed below.

1. The authors introduced a new claim on localization of ORF3 and ORF4 (P-bodies and SGs) to explore possible action mechanism of ORF3-ORF4, particularly of ORF3. Based on the predicted function of

ORF3 as a superfamily II DNA and RNA helicase and the critical role of P-bodies and SGs in RNA metabolism (as referred by authors their deficiency causes postembryonic lethality), does the authors infer that ORF3 destroys P-bodies and SGs (or at least has negative impact on their formation)? I agree this is an interesting insight into ORF3's role as a killer. To support this, I believe it is important to repeat sub-cellular localization of at least ORF3, given the poor (inconvincible) images presented in Supplementary Fig. 16 and different results in the original manuscript. Also, pay attention to any changes of P-bodies and SGs when expressing ORF3 in protoplasts. I recommend using callus or suspension cells to prepare quality protoplasts. The authors did not touch how ORF4 detoxifies ORF3 (they are co-localized but do not interact directly), but I understand that it is a challenging job and could be pursued in future studies.

Response: We deeply appreciate your comments and suggestion. We have repeated the subcellular localization analysis, and have updated images in Supplementary Fig. 17. The localization of ORF3 and ORF4 remains the same as in our last revision, with ORF3 showing signals in both mitochondria and cytoplasmic foci (P-bodies and SGs) and ORF4 being localized in cytoplasmic foci only. The inconsistency in subcellular localization results between the original manuscript and the revised version is due to the different markers we used. In the original manuscript, we used Rf1b as the mitochondrion marker because it was found to be directed to mitochondria in bombarded onion epidermal cells (Wang *et al.*, 2006). Since ORF3 and ORF4 co-localized with Rf1b in our original manuscript and in the Figure below, we made the conclusion that both ORF3 and ORF4 localized in mitochondria in our original manuscript. However, in our last revision, we selected OxCOX11 as the mitochondrion marker (Han *et al.*, 2022) since Rf1b was found to be mainly targeted to cytoplasmic foci

instead of mitochondria (Figure below). In this revision, we further used mitochondrion-specific dye (MitoTracker Deep Red, Invitrogen) to mark mitochondria. We hope that our new data could dispel your doubts.

ORF3 as a superfamily II DNA and RNA helicase coincided with the critical role of P-bodies and SGs in RNA metabolism. Correspondingly, transcriptomic analyses revealed that *pf12A* (allelic to *ORF3*) induced a drastically increased number of differentially expressed genes in the anthers at the mature pollen stage possibly due to aberrant base excision repair and RNA degradation in the gametes (Zhou *et al.*, 2023). However, ORF3 didn't seem to affect the formation of P-bodies and SGs, as the protoplasts of E9 and *orf3* exhibited similar patterns of cytoplasmic foci (Supplementary Fig. 27).

References used in this part of response:

Han *et al.* All-in-one: a robust fluorescent fusion protein vector toolbox for protein localization and BiFC analyses in plants. *Plant Biotechnology Journal*, 20: 1098-1109 (2022).

Wang *et al.* Cytoplasmic male sterility of rice with Boro II cytoplasm is caused by a cytotoxic peptide and is restored by two related PPR motif genes via distinct modes of mRNA silencing. *Plant Cell*, 18: 676-687 (2006).

Zhou *et al.* A minimal genome design to maximally guarantee fertile inter-subspecific hybrid rice. *Molecular Plant*, 16(4): 726-738 (2023).

Figure. Subcellular localization of Rf1b, ORF3 and ORF4.

Rf1b is found to be mainly localized in cytoplasmic foci, and colocalization signal was found between Rf1b and ORF3, and between Rf1b and ORF4.

2. It seems that the authors down-value mitochondrion localization of ORF3 in the revised version. Considering the mitochondrion signal in ORF3 (with localization verified by independent protoplast assays) and the observation that ORF3 inhibits cell growth in yeast, however, the possibility that ORF3 functions in mitochondrion cannot be simply excluded. This concern can be addressed easily by doing complementary transformation with a signal-less version of ORF3. It takes about two months to complete the experiment since transgenic callus (not necessarily transgenic plants) can be used for analysis (to see if high level expression of mutated ORF3 in T65 can be obtained). Alternatively, the base editor system can be used to mutate the signal sequence in the T65/E9 F₁ background to see pollen fertility in mutated plants. This BE approach will take longer time. Either experiment will clarify if mitochondrion localization is essential for functioning of ORF3.

Response: We are grateful for your suggestion. According to your advice, we transformed the overexpression vectors harboring functional *ORF3* (*ORF3*^{Hap-1}), loss-of-function *ORF3*^{HJX74} (*ORF3*^{Hap-2}), *ORF3*^{Δ100} (*ORF3*^{Hap-1} without sequence encoding mitochondrion localization signal), or an empty vector into T65 calli. As expected, the selection frequency of Hyg-resistant callus was extremely low for *ORF3* compared with the empty vector control (Supplementary Fig. 12a-b), supporting our hypothesis that *ORF3* is toxic to living cells without *ORF4* and its expression should be strictly regulated. However, significant amount of Hyg-resistant secondary calli were apparently propagated from *Agrobacterium* co-cultivated primary calli transformed with *ORF3*^{HJX74} and *ORF3*^{Δ100} (Supplementary Fig. 12a-b), and high expression level of transgenes could be obtained in *ORF3*^{Δ100} transgene-positive calli (Supplementary Fig. 12c), supporting that the mitochondrion localization of ORF3 is essential for its function.

We also repeated the yeast transformation analysis including new assay of *ORF3*^{Δ100}. The induction of functional ORF3 in INVSc1 yeast cells significantly inhibited the growth of the transformants, while this growth inhibition was attenuated when *ORF3*^{Δ100} was transformed (Supplementary Fig. 13), further supporting the importance of ORF3's mitochondrion localization on its killer function.

We have revised our description in the Discussion section (Lines 412-428).

3. I agree with Review #3 who gave constructive comments on analysis of evolutionary trajectory of *ORF3-ORF4*. Fig. 5a is confusing to me and I also don't see a clear description (I am not expecting conclusion) on evolution of *ORF3-ORF4*. Surely, the authors made appreciable efforts by analyzing numerous collections of both wild and cultivated rice. Evolution biology is not my field, so I am sorry I cannot give comments for improvement. The authors assume functioning of HAP^{ORF4}-2 in Type-1 *Se*, HAP^{ORF4}-22 to 25 in Type-3 *Se* and HAP^{ORF4}-30 to 34 in Type-4 *Se*, regardless of polymorphisms between them and lacking genetic verification. The authors should do genetic complementation in T65/E9 F₁ background for 2-3 ORF4 haplotypes, particularly HAP^{ORF4}-2 in functional Type-1 *Se*, to verify their assumption.

Response: Thank you for your careful review and comments. We have now heavily revised the whole section on the evolutionary history of *Se* and removed the claim that HAP^{ORF4}-2 (rewritten as ORF4^{Hap-2} in our revision) is functional. We now focus on describing the haplotype diversity of *Se* in *Oryza* species (Fig. 5a and Supplementary Fig. 26). We also performed functional experiments by showing a truncated form of ORF3 (ORF3^{HJX74}) has disrupted gene function (Supplementary Figs. 12-13 and 17). We identified an important functional variant (369-bp InDel) on the promoter of *ORF4* (Supplementary Fig. 23), which we showed affecting the expression

level (Supplementary Fig. 25) but not the subcellular localization (Supplementary Fig. 24) of the gene. We also pointed out that the presence of *Se* locus was restricted to a few *Oryza* AA species, and functional *ORF3* was restricted to Asian rice species complex (Fig. 5a and Supplementary Fig. 22). The *indica* and *japonica* subspecies each harbor different haplotypes with moderate or high frequency. These analyses support our conclusion that the emergence of *Se* locus is associated with the *Oryza* lineage, and the locus is an important player underlying the reproductive barrier between the two subspecies of Asian cultivated rice.

4. Include controls, such as ORF3-AD+BD and ORF4-AD+BD, in Sup. Fig. 17.

Response: Thank you for your careful review. We have included the requested controls, and have updated images in Supplementary Fig. 18.

Other minor points

1. Slightly lower spikelet fertility but statistically significant in T65/E9 F₁ (Fig. 1d), which is consistent with the segregation ratio of T/E and E/E (54:74) from a cross with T/E as female in Supp. Table 2 (although not statistically significant). Those results imply that female gamete carrying T65 allele is slightly affected, but at a negligible degree relative to male gamete. Soften the claim rather than simple argument.

Response: We are grateful for your suggestion. We have softened our claim and revised our description in the Results (Lines 138-143) and Discussion (Lines 537-541) sections.

2. For expression analysis (such as three ORF3 overexpression, two ORF4 overexpression transgenic T₀ plants) and pollen fertility of single T₀

plants (more examples throughout the paper), there is no information on biological replicates in either Method or figure legends. The authors need to indicate how SD was obtained from analysis of a single given plant.

Response: Thank you for your careful review. For rice callus transformation, we normally get more than one seedling from a single positive callus. We usually grow them separately, but use the same name to represent that they originate from the same transgenic event. In our expression analysis of T₀ plants, SD was obtained from the multiple plants generated from the same transgenic event. We have added this description into the Method section (Lines 629-631) in our revised version.

3. Lines 143, change sterility to semi-sterility.
4. Lines 180-182, rephrase the sentence as Fig. 3a shows targets only for ORF3.
5. Supp Table 54 (also Line 204 in Text): typo of Supp Table 5?
6. Line 259, QRT-PCR. qRT-PCR is common.
7. Lines 430-433, soften the sentence (regarding full fertile pollen) as there are other genes controlling *indica/japonica* HMS.
8. Supp. Fig. 5w, the label MP could be BP?

Response: Thank you very much for above corrections and comments. We have revised the manuscript accordingly.

9. Supp. Fig. 13, any copy number information of transgene? It seems the pollen fertility and segregation analysis were single-copy based.

Response: Thank you very much for your careful review. As the pollen fertility of two T₀ ORF4OEs was ~75% (Supplementary Fig. 14b-d), both

transgenic plants must have single-copy transgene, which was similar to the result of *ORF4CTLs* (Fig. 4 and Supplementary Table 7). Therefore, we didn't determine the location of *ORF4OE* transgenes through Tail-PCR as we did for *ORF4CTLs*, but only checked the genotype segregation (*T/T*, *T/E* or *E/E*), and focused on whether *T/T* plants could be obtained in the segregating populations (Supplementary Fig. 14f).

10. Supp. Fig. 26 provides a good example on introgression of the PAV from *indica* to *japonica*. Give interpretation on how “artificial selection” was placed during the breeding procedure, at least in the legend.

Response: Thank you for your advice. Genealogical trees were deduced from the public available data on China Rice Data center (WYG7: <https://www.ricedata.cn/variety/varis/601512.htm>; LG31: <https://www.ricedata.cn/variety/varis/609156.htm>). We have added this information in the figure legend of Supplementary Fig. 28. These two varieties were bred by different breeders in China decades ago. It was unlikely that they artificially selected the PAV region directly. However, both examples showed that the PAV could be introgressed into *japonica* through repeatedly backcrossing *indica* variety with *japonica* varieties, generating rare *japonica* (III) type.

Reviewer #3:

The revised manuscript has tried to address the reviewers' comments, but not yet to a satisfactory degree. I believe that additional experiments and data analysis will strengthen the author's conclusions, with the evolution part in particular. I list a few points below for the authors to consider.

Response: We really appreciate reviewer #2 for evaluating our responses to reviewer #3. We did additional experiments to address the raised questions. Our point-by-point responses are detailed below.

There are several studies placing *O. meridionalis* the earliest in the origin of *Oryza* AA genome. In revised Discussion the authors still insist that *O. longistaminata* is the earliest (again, based on chloroplast analysis). In addition, the FF genome is not addressed. The finding of *ORF3L* and *ORF4L* on chromosome 11 in BB and CC genomes is interesting. I suggest to sequence more accessions for *O. meridionalis*, BB, CC, FF, EE and even GG, to allow the authors to have better understanding on absence and presence of the PAV. Meiotic drivers (like ORF3-ORF4 here) are thought to exist for relatively short evolutionary timespans because a gene driver or gene family is often found in a single species or in a group of very closely related species. Drivers are generally considered doomed to extinction when they spread to fixation or when suppressors arise (due to genomic conflict). Therefore, it is not surprising that this PAV is present in some species but absent in other species. In addition to what the authors proposed (such as translocation), horizontal gene transfer between lineages is also possible (not necessarily through the shared decent from a common ancestor). With those considerations, authors could propose an evolution model on PAV spanning the FF genome and *Oryza sativa*.

Response: Thank you for your careful review and suggestion. According to your advice, we analyzed the PAV in more rice samples of BB-, CC-, FF-, EE-, and GG-genomes. We are sorry that we couldn't sequence more accessions of *O. meridionalis* because no more accessions were available in our lab or from our collaborators.

Using PCR-based assay, we found that in all 4 BB-genome *O. punctata*, 2 CC-genome *O. eichingeri*, 3 CC-genome *O. officinalis*, 3 EE-gnome *O.*

australiensis, 2 FF-genome *O. brachyantha*, and 3 GG-genome *O. meyeriana*, no PAV region was found. Just to note, we previously claimed the existence of putative ORF3 and ORF4 homologs in FF-genome based on BLAST search hitting the published *O. brachyantha* genome which was generated by short-reads sequencing and not with high-quality (Chen et al., 2013). We did not verify whether *ORF3L* and *ORF4L* were present in *O. brachyantha* as a PAV containing *Se*, and it is likely that our BLAST search hit another paralogs. To rigorously address the presence and absence of *Se* locus, we PCR assayed two *O. brachyantha* accessions and showed that the PAV was absent, which is consistent with the results that it is also absent in *Oryza* species of BB-, CC-, EE- and GG-genome. Therefore, the PAV is only found to be polymorphic in AA-genome.

We have now revised the section on the evolutionary history of *Se* and removed claims that were not well supported by our data. We now focus on describing the haplotype diversity of *Se* in *Oryza* species (Fig. 5a and Supplementary Fig. 26). We also performed functional experiments by showing a truncated form of ORF3 (ORF3^{HJX74}) has disrupted gene function (Supplementary Figs. 12-13 and 17). We identified an important functional variant (369-bp InDel) on the promoter of *ORF4* (Supplementary Fig. 23), which we showed affecting the expression level (Supplementary Fig. 25) but not the subcellular localization (Supplementary Fig. 24) of the gene. We also pointed out that the presence of *Se* locus was restricted to a few *Oryza* AA species, and functional ORF3 was restricted to Asian rice species complex (Fig. 5a and Supplementary Fig. 22). The *indica* and *japonica* subspecies each harbor different haplotypes with moderate or high frequency. These analyses support our conclusion that the emergence of *Se* locus is associated with the *Oryza* lineage, and the locus is an important player underlying the reproductive barrier between the two subspecies of Asian cultivated rice.

Reference used in this part of response:

Chen *et al.* Whole-genome sequencing of *Oryza brachyantha* reveals mechanisms underlying *Oryza* genome evolution. *Nature Communications*, 4: 1595 (2013).

The reviewer likes to see experimental (direct) evidence to support the claim that the PAV-mediated reproductive isolation exists in wild progenitors. Clearly, the authors didn't address this comment by performing experiments, and not explain why not doing so (e.g., time and technical issues). Instead, the authors emphasize the claim by inferring and deducing the existing data. It is understood that genetic experiments take time. I suggest quick transient assays to see subcellular localization of ORF4 Hap^{se}-2 (better if include more representing haplotypes) to infer their functionality.

Response: Thank you for your suggestion. We think our claim that PAV-mediated reproductive isolation exists in wild progenitors is well supported as the *Se* locus is polymorphic within the species: 235/342 individuals of *O. rufipogon* have the locus deleted, while 19/342 individuals harbor Type-1 *Se*.

In addition, we checked the subcellular localization of several ORF4 haplotypes according to your advice. As shown in Supplementary Fig. 24, different ORF4 haplotypes (including ORF4^{Hap-2}) showed the similar localization in cytoplasmic foci, suggesting the functionality of all these full-length ORF4s.

One suggestion: highlight modern *japonica* and *indica* varieties in S. Table 13 (40 and 19, respectively, as mentioned in the above response).

Response: Thank you for the suggestion. We have revised Supplementary Table 13 accordingly.

ORF3 functioning as a killer is not questionable. The reviewer likes to see direct evidence on association of low expression of *ORF3* and failure of complementary experiments. The authors can perform Western experiments on those low-expression plants. Alternatively, the authors can knock-down *ORF3* expression by using RNAi technology in T65/E9 F₁ to see relationship between reduced *ORF3* expression and restoration of pollen fertility. Another suggestion is to use an inducible promoter to drive functional *ORF3*^{E9} (Hap^{*ORF3*}-1) and loss-of-function *ORF3*^{HJX74} (Hap^{*ORF3*}-2, as a control) in T65 callus to see retardation of callus growth once *ORF3* expression is triggered on.

Response: We are grateful for your suggestion. We are sorry that we can't perform western blot experiments as we do not have antibody of *ORF3* and we didn't add a protein tag to *ORF3* in our complementation experiments. According to your advice, we transformed the overexpression vector harboring functional *ORF3* (*ORF3*^{Hap-1}), loss-of-function *ORF3*^{HJX74} (*ORF3*^{Hap-2}), *ORF3*^{Δ100} (encoding *ORF3* without mitochondrion localization signal), or an empty vector into T65 calli. Significant amount of Hyg-resistant secondary calli were apparently propagated from *Agrobacterium* co-cultivated primary calli transformed with *ORF3*^{HJX74}, *ORF3*^{Δ100}, or an empty vector (Supplementary Fig. 12a-b), and high level expression of transgenes could be obtained (Supplementary Fig. 12c). However, the selection frequency of Hyg-resistant callus and the expression of positive transgene was extremely low when *ORF3* was transformed (Supplementary Fig. 12), directly supporting our hypothesis that *ORF3* is toxic to living cells without *ORF4* and its expression should be strictly regulated.

In addition, we also repeated the yeast transformation analysis, and the induction of functional *ORF3* significantly inhibited the growth of the

transformants (Supplementary Fig. 13), further supporting the killer function of ORF3.

Reviewers' Comments:

Reviewer #2:

Remarks to the Author:

The authors have added new experimental data that provide more convincing supports to the relevant claims, and revised the manuscript accordingly. All comments/concerns from Reiewer #3 and me have been well addressed. I would like to recommend publication of this study in NC.

Response to reviewer's comments:

Reviewer #2:

The authors have added new experimental data that provide more convincing supports to the relevant claims, and revised the manuscript accordingly. All comments/concerns from Reiewer #3 and me have been well addressed. I would like to recommend publication of this study in NC.

Response: Thank you for your time. We really appreciate your efforts in reviewing our manuscript.